# PACIA: Parameter-Efficient Adapter for Few-Shot Molecular Property Prediction

## Abstract

Molecular property prediction (MPP) plays a crucial role in biomedical applications, but it often encounters challenges due to a scarcity of labeled data. Existing works commonly adopt gradient-based strategy to update a large amount of parameter for property-level adaptation. However, the increase of adaptive parameters can cause overfitting and lead to poor performance. Observing that graph neural network (GNN) performs well as both encoder and predictor, we propose PACIA, a parameter-efficient GNN adapter for few-shot MPP. We design a unified adapter to generate a few adaptive parameters to modulate the message passing process of GNN. We then adopt hierarchical adaptation mechanism to adapt the encoder on property-level and the predictor on molecule-level by the unified GNN adapter. Extensive results show that PACIA obtains the state-of-the-art performance in few-shot MPP problems, and our proposed hierarchical adaptation mechanism is rational and effective.

## 1 Introduction

Molecular property prediction (MPP) (Wieder et al., 2020) which predicts whether desired properties will be active on given molecules, can be naturally modeled as a few-shot learning problem (Waring et al., 2015; Altae-Tran et al., 2017). As wet-lab experiments to evaluate the actual properties of molecules are expensive and risky, usually only few labeled molecules are available for certain property. While recently, Graph Neural Networks (GNN) are popularly used to learn molecular representations (Xu et al., 2019; Yang et al., 2019; Xiong et al., 2019). Modeling molecules as graphs, GNN can capture inherent structural information. Hence, GNN-based methods obtain better performance than classical ones (Unterthiner et al., 2014; Ma et al., 2015), especially when they are pretrained on self-learning tasks. As for tasks with only a few labeled molecules, the performance of existing GNN-based methods is still far from desired.

Various few-shot learning (FSL) methods have been developed to handle few-shot MPP problem. The earlier work IterRefLSTM (Altae-Tran et al., 2017) builds a metric-based model upon matching network (Vinyals et al., 2016). Subsequent works mainly adopt gradient-based meta learning strategy (Finn et al., 2017) to handle the standard few-shot MPP problem, which learns parameter initialization with good generalizability across different properties and adapts parameters by gradient descent for target property. Specifically, Meta-MGNN (Guo et al., 2021) brings chemical prior knowledge in the form of molecular reconstruction loss, and optimizes all parameters by gradient descents. PAR (Wang et al., 2021) introduces attention and relation graph module to better utilize the labeled samples for property-adaptation with the awareness of target chemical property, and conducts a selective gradient-based meta learning strategy. ADKF-IFT (Chen et al., 2022) takes a gradient-based meta learning strategy with implicit function theorem to avoid computing expensive hypergradients, and builds a Gaussian Process for each task as classifier. There are also works that bring auxiliary information such as additional reference molecules from large molecule database (Schimunek et al., 2023) and auxiliary properties (Zhuang et al., 2023) to improve the performance of few-shot MPP (Schimunek et al., 2023; Zhuang et al., 2023).

There are two main problems in existing works. First, they ignore molecule-level difference. The chemical space is enormous and the representations of molecules vary in a wide range. The molecule-level difference should be addressed in few-shot MPP when classifying the encoded molecules. When molecules are more similar to the labeled molecules in one certain class, they can be easily classified. While others exhibit comparable similarity to both categories, they will be harder to

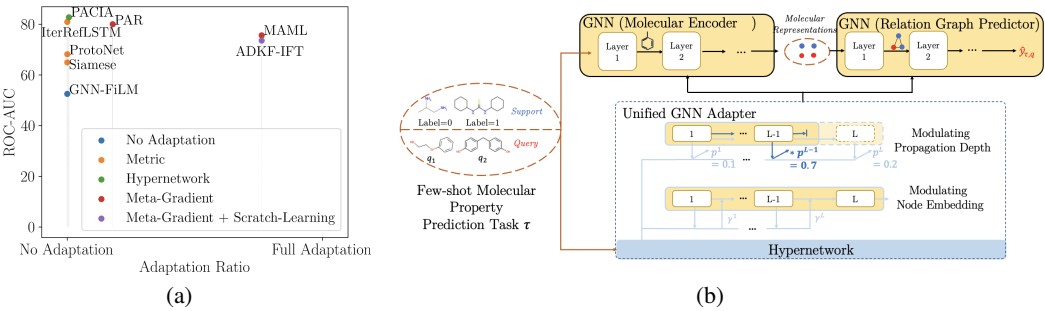

(a)                  (b)

Figure 1: Illustration of the proposed PACIA. (a), The adaptation ratio ($\frac{|adaptive\ params|}{|total\ params|}$) of different methods and their testing performance on 1-shot tasks of Tox21. (b), Under the encoder-predictor framework in few-shot MPP methods, PACIA use a unified GNN adapter, where hypernetwork generates adaptive parameters (marked in blue) to modulate node embeddings and propagation depths in both GNN encoder and predictor.

classify accurately. Thus, a fixed predictor can not fit all molecules even in a single task. Second, gradient-based meta learning strategy requires updating a large number of parameters to adapt to each few-shot task. This results in poor learning efficiency and is prone to overfit given insufficient labeled samples (Rajeswaran et al., 2019; Yin et al., 2020), as shown in Figure 1 (a). With more task-specific parameters, the model gets easier to overfit, which would be more severe in extreme few-shot cases. Effective adaptation should be made in a parameter-efficient way, i.e, without modulating a large amount of parameters.

In this paper, we propose **PACIA**, a **PA**rameter-effi**CI**ent **A**dapter for few-shot MPP problem. PACIA applies hierarchical adaptation consisting of property-level adaptation which aims to obtain property-adaptive molecular representations, and molecule-level adaptation which further adjust predictor to be molecule-adaptive. The adapter is designed upon hypernetworks (Ha et al., 2017), which are neural networks learned to generate parameters of the main network. In particular, we first summarize existing works into encoder-predictor framework, where GNN performs well acting as both encoder and predictor. Then, we design a unified GNN adapter to generate a few adaptive parameters to modulate message passing process of GNN in two aspects: node embedding and propagation depth. Consequently, the encoder is adapted on property-level and the predictor is adapted on molecule-level. During learning, the parameters of hypernetwork and the main network (including encoder and predictor) are meta-learned across tasks to capture shared knowledge. While the task-specific and molecule-specific knowledge are captured by a few adaptive parameters generated via a single forward pass of hypernetworks without taking optimization steps. In this way, the risk of overfitting is alleviated (Figure 1 (a)). Extensive results show PACIA obtains the state-of-the-art performance on benchmark few-shot MPP datasets MoleculeNet Wu et al. (2018) and FS-Mol Stanley et al. (2021). We also closely examine and validate the effectiveness of our hierarchical adaptation mechanism.

## 2 RELATED WORKS

**Few-Shot Learning.** Few-shot learning (FSL) aims to generalize to a task with a few labeled samples (Wang et al., 2020). In terms of adaptation mechanism, existing FSL methods can be classified into three main categories: (i) gradient-based approaches (Finn et al., 2017; Grant et al., 2018) which learn the model or optimizer that can fast optimize the parameter for new task, (ii) metric-based approaches (Vinyals et al., 2016; Snell et al., 2017) which learn a model learn the embedding function and metric where correct relations can be built between samples, and (iii) amortization-based approaches (Requeima et al., 2019; Lin et al., 2021; Przewiezlikowski et al., 2022) which uses hypernetworks to map the labeled samples in the task to a few parameters to adjust the main networks to be task-specific. Recent works (Requeima et al., 2019) found that amortization-based approaches can reduce overfitting compared with gradient-based methods. They also have faster inference speed as the adapted parameters are generated by a single forward pass without taking optimization steps. Besides, the main networks can approximate various functions in addition to distance-based ones.

**Hypernetworks.** Hypernetworks (Ha et al., 2017) refer to neural networks which learn to generate parameters for another neural network. The main network learns to map some raw inputs to their desired targets, whereas another group of inputs are fed to the hypernetwork to generate parameters to adapt the main network. It has been successfully used to handle various applications like cold-

start recommendation (Lin et al., 2021) and image classification (Przewiezlikowski et al., 2022) Designing appropriate hypernetworks is challenging, requiring domain knowledge to decide what information to be fed into hypernetworks, how to adapt the main network, and what is the appropriate architecture of hypernetworks. There exist hypernetworks for general GNNs. Brockschmidt (2020) builds hypernetworks taking target node as input to generate parameters to modulate weight matrix in aggregation function in message passing. And Nachmani & Wolf (2020) design a hypernetwork for node-specific message passing functions that can lead to a boost in performance. In contrast to them, we particularly consider designing parameter-efficient modulators for GNNs used in encoder-predictor framework for few-shot MPP.

## 3 PRELIMINARIES OF FEW-SHOT MOLECULAR PROPERTY PREDICTION

### 3.1 PROBLEM SETUP

In a few-shot MPP task $\mathcal{T}_\tau$, each sample $\mathcal{X}_{\tau,i}$ is a molecular graph and its label $y_{\tau,i} \in \{0, 1\}$ records whether the molecule is inactive or active on a certain property. Only a few labeled samples are available in the task. Following earlier works (Altae-Tran et al., 2017; Stanley et al., 2021; Chen et al., 2022; Schimunek et al., 2023), we model a few-shot MPP task as a 2-way classification task $\mathcal{T}_\tau$, where a support set $\mathcal{S}_\tau = \{(\mathcal{X}_{\tau,s}, y_{\tau,s})\}_{s=1}^{N_\tau}$ contains labeled samples from inactive/active class and a query set $\mathcal{Q}_\tau = \{(\mathcal{X}_{\tau,q}, y_{\tau,q})\}_{q=1}^{M_\tau}$ contains $M_\tau$ samples whose labels are only used for evaluation. Note that this work covers both settings including balanced support sets, i.e., $\mathcal{S}_\tau$ contains $\frac{N_\tau}{2}$ samples per class which is consistent with the standard $N$-way $K$-shot FSL setting (Altae-Tran et al., 2017), and imbalanced support sets which exist in real-world applications (Stanley et al., 2021). This work aims at learning a model from a set of tasks $\{\mathcal{T}_\tau\}_{j=1}^N$ that can generalize to new task given the support set. Specifically, the target properties are different across tasks.

### 3.2 ENCODER-PREDICTOR FRAMEWORK

Existing works adopt an encoder-predictor framework to solve the (few-shot) MPP problem. In the past, molecules are encoded with certain properties (fingerprint vectors (Rogers & Hahn, 2010)) and fed to deep networks for prediction (Unterthiner et al., 2014; Ma et al., 2015). While recently, GNNs are popularly taken as molecular encoders (Li et al., 2018; Yang et al., 2019; Xiong et al., 2019; Hu et al., 2019) due to their superior performance on learning from of topological data.

Given molecular graphs which are graphs of atoms connected by chemical bonds, a GNN encoder maps them to molecular representations which are vectors with fixed length. Consider a molecular graph $\mathcal{X} = \{\mathcal{V}, \mathcal{E}\}$ with node feature $\mathbf{h}_v$ for each atom $v \in \mathcal{V}$ and edge feature $\mathbf{b}_{vu}$ for each chemical bond $e_{vu} \in \mathcal{E}$ between atoms $v, u$. At the $l$th layer, GNN updates atom embedding $\mathbf{h}_v^l$ of $v$ as

$$\mathbf{h}_v^l = \text{UPDATE}^l\left(\mathbf{h}_v^{l-1}, \text{AGG}^l\left(\{(\mathbf{h}_v^{l-1}, \mathbf{h}_u^{l-1}, \mathbf{b}_{vu}) | u \in \mathcal{H}(v)\}\right)\right), \tag{1}$$

where $\mathcal{H}(v)$ contains neighbors of $v$. $\text{AGG}(\cdot)$ and $\text{UPDATE}(\cdot)$ are aggregation and updating functions respectively. After $L$ layers, the molecule-level representation $\mathbf{r}$ for $\mathcal{X}$ is obtained as

$$\mathbf{r} = \text{READOUT}\left(\{\mathbf{h}_v^L | v \in \mathcal{V}\}\right), \tag{2}$$

where $\text{READOUT}(\cdot)$ function aggregates all atom embeddings. The encoder gets molecular representation $\mathbf{r}_{\tau,i}$ for each molecule $\mathcal{X}_{\tau,i}$, which is then fed to the predictor for classification.

Then, a predictor $f(\cdot)$ assigns label for a query molecule $\mathcal{X}_{\tau,q}$ given support molecules in $\mathcal{S}_\tau$:

$$\hat{\mathbf{y}}_{\tau,q} = f(\mathbf{r}_{\tau,q} | \{\mathbf{r}_{\tau,s}\}_{s \in \mathcal{S}_\tau}). \tag{3}$$

The specific choice of $f(\cdot)$ is diverse, e.g., pair-wise similarity (Altae-Tran et al., 2017), multi-layer perceptron (MLP) (Guo et al., 2021; Wang et al., 2021) and Mahalanobis distance (Stanley et al., 2021). Recently Wang et al. (2021) found that learning with relation graphs can effectively compensate for the lack of supervised information. In particular, the molecular representations are refined on relation graphs such that the similar molecules cluster closer. Initialize molecular representations as the output of the encoder, i.e., $\mathbf{h}_{\tau,i}^0 = \mathbf{r}_{\tau,i}$. Denote the set of $N_\tau + 1$ molecules as $\mathcal{R}_{\tau,q} = (\mathcal{X}_{\tau,q}, y_{\tau,q}) \cup \mathcal{S}_\tau$, which contains all information to make prediction for query $q$. The relation graph works by recurrently estimating the adjacency matrix and updating the molecular

representations. At the $l$th layer, each element $a_{ij}^l$ in the adjacent matrix $\mathbf{A}_{\tau,q}^l$ of the relation graph is learned to represent pair-wise similarities between any two molecules in $\mathcal{R}_{\tau,q}$:

$$a_{ij}^l = \begin{cases} \text{MLP}\left(|\mathbf{h}_{\tau,i}^{l-1} - \mathbf{h}_{\tau,j}^{l-1}|\right) & \text{if } i \neq j \\ 1 & \text{otherwise} \end{cases}. \tag{4}$$

Then, each molecular representation is refined as

$$\mathbf{h}_{\tau,i}^l = \text{MLP}(\sum\nolimits_{j=1}^{N_\tau+1} a_{ij}^l \mathbf{h}_{\tau,j}^{l-1}). \tag{5}$$

After $L$ layers of refinement, $\mathbf{h}_{\tau,q}^L$ and $\mathbf{h}_{\tau,s}^L$ (in place of $\mathbf{r}_{\tau,q}$ and $\mathbf{r}_{\tau,s}$) are fed to (3) to obtain final prediction $\hat{\boldsymbol{y}}_{\tau,q}$ for query molecule $\mathcal{X}_\tau$.

## 4 HIERARCHICAL ADAPTATION OF ENCODER-PREDICTOR FRAMEWORK

To generalize across different tasks with a few labeled molecules, existing works (Wang et al., 2021; Chen et al., 2022) usually conduct property-level adaptation by gradient-based meta learning (see Appendix B). However, as discussed in Section 2, gradient-based meta learning optimizes most parameters by a limited amount of supervised information, which is slow to optimize and easy to overfit. As for molecule-level adaptation, gradient is not accessible for each query molecule in testing. Therefore, we instead turn to hypernetworks to achieve parameter-efficient adaptation.

Our proposed PACIA is shown in Figure 1, and we provide a more detailed figure in Appendix D. As both encoder and predictor introduced in Section 3.2 are based on GNN, we design a unified GNN adapter to generate a few adaptive parameters to hierarchically adapt the encoder on property-level and the predictor on molecule-level in a parameter-efficient manner. Next, we first introduce the unified GNN adapter (Section 4.1), then describe how to learn the main networks (including encoder and decoder) with the unified GNN adapter by episodic training (Section 4.2).

### 4.1 A UNIFIED GNN ADAPTER

To adapt GNN's parameter-efficiently, we design a GNN adapter to modulate the node embedding and propagation depth, which are essential in message passing process.

**Modulating Node Embedding.** Denote the node embedding at the $l$th layer as $\mathbf{h}^l$, which can be atom embedding in encoder or molecular embedding in relation graph predictor. We obtain adapted embedding $\hat{\mathbf{h}}^l$ as

$$\hat{\mathbf{h}}^l = e(\mathbf{h}^l, \boldsymbol{\gamma}^l), \tag{6}$$

where $e(\cdot)$ is an element-wise function, and $\boldsymbol{\gamma}^l$ is adaptive parameter generated by the hypernetwork. This adapted embedding $\hat{\mathbf{h}}^l$ is then fed to next layer of message passing.

**Modulating Propagation Depth.** Further, we manage to modulate the propagation depth, i.e., layer number $l$ of a GNN. Controlling $l$ is challenging since it is discrete. We achieve this by training a differentiable controller, which is a hypernetwork to generate a scalar $p^l$ corresponding to each $\{l\}_{l=1}^L$ where $L$ is the maximum layer number. The value of $p^l$ estimates how likely the message passing should stop after layer $l$. The hypernetwork is shared across all $L$ layers. Finally, as there are $L$ layers in total, the vector

$$\boldsymbol{p} = \text{softmax}([\, p^1, \; p^2, \; \cdots, \; p^L \,]), \tag{7}$$

represents the plausibility of choosing each layer. During meta-training, $\boldsymbol{p}$ is used as adaptive parameter to modulate GNN layers, so that gradients can be propagated to parameter in the hypernetwork to generate $p$. Specifically, after propagation through all $L$ layers, the final embedding of each node is

$$\widetilde{\mathbf{h}} = \sum\nolimits_{l=1}^L [\boldsymbol{p}]_l \mathbf{h}^l, \tag{8}$$

where $[\boldsymbol{p}]_l$ is the $l$th element of $\boldsymbol{p}$.

**Generating Adaptive Parameters by Hypernetwork.** We use hypernetworks to generate adaptive parameters $\{\boldsymbol{\gamma}^l, p^l\}_{l=1}^L$. In particular, note that the generated adaptive parameter should be permutation-invariant to the order of input samples in $\mathcal{S}_\tau$. Therefore, we first calculate class prototypes $\boldsymbol{r}_{\tau,+}^l$ and $\boldsymbol{r}_{\tau,-}^l$ of active class (+) and inactive class (-) for samples in $\mathcal{S}_\tau$ by

$$\begin{aligned} \boldsymbol{r}_{\tau,+}^l &= \frac{1}{|\mathcal{S}_\tau^+||\mathcal{V}_{\tau,s}|} \sum\nolimits_{\mathcal{X}_{\tau,s} \in \mathcal{S}_\tau^+} \text{MLP}([\sum\nolimits_{v \in \mathcal{X}_{\tau,s}} \mathbf{h}_v^l \mid \boldsymbol{y}_{\tau,s}]), \\ \boldsymbol{r}_{\tau,-}^l &= \frac{1}{|\mathcal{S}_\tau^-||\mathcal{V}_{\tau,s}|} \sum\nolimits_{\mathcal{X}_{\tau,s} \in \mathcal{S}_\tau^-} \text{MLP}([\sum\nolimits_{v \in \mathcal{X}_{\tau,s}} \mathbf{h}_v^l \mid \boldsymbol{y}_{\tau,s}]), \end{aligned} \tag{9}$$

where $[\cdot|\cdot]$ means concatenating, $\mathcal{S}_\tau^+$ and $\mathcal{S}_\tau^-$ are the sets of active and inactive samples in $\mathcal{S}_\tau$, and $\boldsymbol{y}_{\tau,s}$ is the one-hot encoding of label. Using $\boldsymbol{r}_{\tau,+}^l$ and $\boldsymbol{r}_{\tau,-}^l$ allows subsequent steps to keep supervised information while being permutation-invariant.

For property-level adaptation, we then map $\boldsymbol{r}_{\tau,+}^l$ and $\boldsymbol{r}_{\tau,-}^l$ to $\{\boldsymbol{\gamma}_\tau^l, p_\tau^l\}_{l=1}^L$ as

$$[\boldsymbol{\gamma}_\tau^l, p_\tau^l] = \mathtt{MLP}\left([\boldsymbol{r}_{\tau,+}^l \mid \boldsymbol{r}_{\tau,-}^l]\right). \tag{10}$$

As for molecule-level adaptation, information comes from both $\mathcal{S}_\tau$ and the specific query molecule $q$. Likewise, we use class prototypes $\boldsymbol{r}_{\tau,+}^l$ and $\boldsymbol{r}_{\tau,-}^l$ to keep permutation-invariant. We then generate $\{\boldsymbol{\gamma}_{\tau,q}^l, p_{\tau,q}^l\}_{l=1}^L$ as

$$[\boldsymbol{\gamma}_{\tau,q}^l, p_{\tau,q}^l] = \mathtt{MLP}\left([\mathbf{r}_{\tau,+}^l \mid \mathbf{r}_{\tau,-}^l \mid \sum_{v \in \mathcal{X}_{\tau,q}} \mathbf{h}_v^l]\right), \tag{11}$$

where molecule-specific information in $\mathcal{X}_{\tau,q}$ is considered together with $\mathcal{S}_\tau$. Note that parameters of these $\mathtt{MLP}$s in hypernetwork are meta-learned together with the encoder and predictor.

## 4.2 LEARNING AND INFERENCE

Denote the collection of all model parameters in main network((1)-(5)) and hypernetwork ((9)-(11)) as $\boldsymbol{\Theta}$, excluding adaptive parameters. The objective is to minimize

$$\min \sum_{\tau=1}^N \mathcal{L}_\tau, \text{ where } \mathcal{L}_\tau = -\sum_{\boldsymbol{x}_{\tau,q} \in \mathcal{Q}_\tau} \boldsymbol{y}_{\tau,q}^\top \log\left(\hat{\boldsymbol{y}}_{\tau,q}\right). \tag{12}$$

$\mathcal{L}_\tau$ is the loss in task $\mathcal{T}_\tau$, $\boldsymbol{y}_{\tau,q}$ is one-hot true label vector and $\hat{\boldsymbol{y}}_{\tau,q}$ is prediction obtained by (3).

---

**Algorithm 1** Meta-training in PACIA.

**Input:** meta-training task set $\boldsymbol{\mathcal{T}}_{\text{train}}$
1: initialize $\boldsymbol{\Theta}$ randomly or use a pretrained one;
2: **while** not done **do**
3:    **for** each task $\mathcal{T}_\tau \in \boldsymbol{\mathcal{T}}_{\text{train}}$ **do**
4:       **for** $l \in \{1, 2, \cdots, L_{\text{enc}}\}$ **do**
5:          + generate $[\boldsymbol{\gamma}_\tau^l, p_\tau^l]$ by (10);
6:          modulate atom embedding $\mathbf{h}_v^l \leftarrow e(\mathbf{h}_v^l, \boldsymbol{\gamma}_\tau^l)$;
7:          * update atom embedding $\mathbf{h}_v^l$ by (1);
8:       **end for**
9:       obtain atom embedding after message passing $\mathbf{h}_v^{L_{\text{enc}}} \leftarrow \sum_{l=1}^{L_{\text{enc}}}[\boldsymbol{p}_\tau]_l \mathbf{h}_v^l$ and obtain molecular embeddings by (2);
10:    **end for**
11:    **for** each query $(\mathcal{X}_{\tau,q}, y_{\tau,q}) \in \mathcal{Q}_\tau$ **do**
12:       **for** $l \in \{1, 2, \cdots, L_{\text{rel}}\}$ **do**
13:          + generate $[\boldsymbol{\gamma}_{\tau,q}^l, p_{\tau,q}^l]$s by (11);
14:          modulate molecular embedding $\mathbf{h}_{\tau,i}^l \leftarrow e(\mathbf{h}_{\tau,i}^l \boldsymbol{\gamma}_{\tau,q}^l)$;
15:          * update molecular embedding by (4)-(5);
16:       **end for**
17:       obtain molecular embedding after message passing $\mathbf{h}_{\tau,i}^{L_{\text{rel}}} \leftarrow \sum_{l=1}^{L_{\text{rel}}}[\boldsymbol{p}_{\tau,q}]_l \mathbf{h}_{\tau,i}^l$;
18:       obtain prediction $\hat{\mathbf{y}}_{\tau,q}$ by (3);
19:    **end for**
20:    calculate loss by (12)
21:    update $\boldsymbol{\Theta}$ by gradient descent;
22: **end while**
23: **return** learned $\boldsymbol{\Theta}^*$.

---

**Algorithm 2** Meta-testing in PACIA.

**Input:** learned $\boldsymbol{\Theta}^*$, a meta-testing task $\mathcal{T}_\tau$;
1: **for** each task $\mathcal{T}_\tau \in \boldsymbol{\mathcal{T}}_{\text{train}}$ **do**
2:    **for** $l \in \{1, 2, \cdots, L_{\text{enc}}\}$ **do**
3:       + generate $[\boldsymbol{\gamma}_\tau^l, p_\tau^l]$ by (10);
4:       modulate atom embedding $\mathbf{h}_v^l \leftarrow e(\mathbf{h}_v^l, \boldsymbol{\gamma}_\tau^l)$;
5:       * update atom embedding $\mathbf{h}_v^l$ by (1);
6:    **end for**
7:    select propagation depth by (13) and obtain atom embedding after message passing $\mathbf{h}_v^{L_{\text{enc}}} \leftarrow \mathbf{h}_v^{l'}$;
8: **end for**
9: *obtain molecular embedding by (2);
10: **for** each query $(\mathcal{X}_{\tau,q}, y_{\tau,q}) \in \mathcal{Q}_\tau$ **do**
11:    **for** $l \in \{1, 2, \cdots, L_{\text{rel}}\}$ **do**
12:       + generate $[\boldsymbol{\gamma}_{\tau,q}^l, p_{\tau,q}^l]$s by (11);
13:       modulate molecular embedding $\mathbf{h}_{\tau,i}^l \leftarrow e(\mathbf{h}_{\tau,i}^l \boldsymbol{\gamma}_{\tau,q}^l)$;
14:       * update molecular embeddings by (4)-(5);
15:    **end for**
16:    select propagation depth by (13) and obtain molecular embedding after message passing $\mathbf{h}_{\tau,i}^{L_{\text{rel}}} \leftarrow \mathbf{h}_{\tau,i}^{l'}$;
17:    obtain prediction $\hat{\mathbf{y}}_{\tau,q}$ by (3).
18: **end for**

---

"*" (resp. "+") indicates the line is executed by the main network (resp. hypernetwork).

Algorithm 1 summarizes the training procedure of PACIA. As mentioned above, our unified GNN adapter can modulate the node embedding and propagation depth simultaneously, and we cascade the adaption in encoder and predictor. During training, molecular graphs $\mathcal{X}_{\tau,i}$ are first processed by

encoder (line 4-9). At each layer, adaptive parameters $[\boldsymbol{\gamma}_\tau^l, p_\tau^l]$ are obtained by (10) (line 5). Then, (6) immediately modulates all atom embeddings $\mathbf{h}_v^l$ (line 6). After $L_{\text{enc}}$ iterations of message passing (1), (8) is applied before (2), to get property-adaptive molecular representations and initialize node embeddings in relation graph $\mathbf{h}_{\tau,i}^0 = \mathbf{r}_{\tau,i}$ (line 9). Then in predictor, at each layer of GNN on relation graph, adaptive parameters $[\boldsymbol{\gamma}_{\tau,q}^l, p_{\tau,q}^l]$ are obtained with (11) (line 13) and (6) immediately functions on all node embeddings $\mathbf{h}_{\tau,i}^l$ (line 14). After $L_{\text{rel}}$ iterations of message passing (4)(5) and modulation, (8) is applied (line 17). And the final prediction $\hat{\mathbf{y}}_{\tau,q}$ is obtained by (3).

Algorithm 2 provides the testing procedure. The process is similar. The node embeddings are adapted by (6). The propagation depth are adapted by selecting the layer with maximal plausibility (line 7 and 16):

$$l' = \text{argmax}_{l \in \{1,2,\cdots,L\}} \ p^l. \tag{13}$$

Then, node embeddings $\mathbf{h}^{l'}$ are fed forward to the classifier (3).

**Discussion.** In PACIA, model parameter $\boldsymbol{\Theta}$ is shared across all tasks, and the adaptive parameter $\{\boldsymbol{\gamma}^l\}_{l=1}^L$ in (6) and $\{p^l\}_{l=1}^L$ in (7) are generated by hypernetworks. The size of adaptive parameter is far smaller than the main network. This realizes parameter-efficient adaptation and mitigates the risk of overfitting. Also, a pre-trained GNN can be incorporated in our method, which can provide a better starting point to obtain better performance. More discussion about comparison with existing works is provided in Appendix B.1.

## 5 Experiments

In this section, we evaluate the proposed PACIA for few-shot molecular property prediction problems. We run all experiments with 10 random seeds, and report the mean and standard deviations. Appendix C provides more information of datasets, baselines, and implementation details.

### 5.1 Performance Comparison on MoleculeNet

**Setting.** We first conduct experiments on Tox21 (National Center for Advancing Translational Sciences, 2017), SIDER (Kuhn et al., 2016), MUV (Rohrer & Baumann, 2009) and ToxCast (Richard et al., 2016) from MoleculeNet (Wu et al., 2018), which are commonly used to evaluate the performance on few-shot MPP (Altae-Tran et al., 2017; Wang et al., 2021). We adopt the public data split provided by (Wang et al., 2021). The support sets are balanced, each of them contains $K$-shot per class, where $K = 1$ and $K = 10$ are considered. The performance is evaluated by ROC-AUC calculated on the query set of each meta-testing task and averaged across meta-testing tasks.

**Baselines.** We compare with the following baselines: 1) single-task methods: **Random Forest**, and **GNN-ST** (Gilmer et al., 2017); 2) multi-task pretraining methods: **GNN-MT** (Corso et al., 2020; Gilmer et al., 2017); 3) self-supervised pretraining methods: **MAT** (Maziarka et al., 2020); and 4) meta-learning methods:**Siamese** (Koch et al., 2015), **ProtoNet** (Snell et al., 2017), **MAML** (Finn et al., 2017), **EGNN** (Kim et al., 2019). 5) methods proposed for few-shot MPP, including **IterRefLSTM** (Altae-Tran et al., 2017), **PAR** (Wang et al., 2021) and **ADKF-IFT** (Chen et al., 2022). Note that **MHNfs** (Schimunek et al., 2023) is not included as it uses additional reference molecules from external datasets, which leads to unfair comparison. **GS-META** (Zhuang et al., 2023) has not been compared since that approach requires multiple properties of each molecule, which would be not applicable when a molecule is accessible to only one property. Following earlier works Guo et al. (2021); Wang et al. (2021), we use GIN (Xu et al., 2019) as encoder, which is trained from scratch. We also provide results obtained with a pretrained GIN in Appendix C.3.

**Performance.** Table 1 shows the results. Results of Siamese and IterRefLSTM are copied from (Altae-Tran et al., 2017) as their codes are unavailable, and their results on ToxCast are unknown. GNN-FiLM is designed as a GNN model rather than for few-shot MPP, which explains its bad performance. TheName obtains the highest ROC-AUC scores on all cases except the 10-shot case on MUV, where ADKF-IFT outperforms the others by a large margin. This can be a special case where ADKF-IFT works very well but may not be generalizable. Moreover, depending on the number of local-update steps of ADKF-IFT, PACIA is about 5 times faster than ADKF-IFT (both meta-training and inference time is about 1/5). In terms of average performance, PACIA significantly

outperforms the second-best method ADKF-IFT by 3.25%. Results in Appendix C shows that methods with pretrained encoder exhibit similar performance, where our PACIA with pretrained encoder (Pre-PACIA) performs the best. Results in Appendix C.6 shows the performance in a wider range of support set size.

Table 1: Test ROC-AUC obtained on MoleculeNet. The best results are bolded, second-best results are underlined.

| Method | Tox21 | | SIDER | | MUV | | ToxCast | |
|---|---|---|---|---|---|---|---|---|
| | 10-shot | 1-shot | 10-shot | 1-shot | 10-shot | 1-shot | 10-shot | 1-shot |
| Random Forest | - | - | - | - | - | - | - | - |
| GNN-ST | $61.23_{(0.89)}$ | $55.49_{(2.31)}$ | $56.25_{(1.50)}$ | $52.98_{(2.12)}$ | $54.26_{(3.61)}$ | $51.42_{(5.11)}$ | $55.66_{(1.47)}$ | $51.80_{(1.99)}$ |
| MAT | $64.84_{(0.93)}$ | $54.90_{(1.89)}$ | $57.45_{(1.26)}$ | $52.97_{(3.00)}$ | $56.19_{(2.88)}$ | $52.01_{(4.05)}$ | $58.50_{(1.62)}$ | $52.41_{(2.34)}$ |
| GNN-MT | $69.56_{(1.10)}$ | $62.08_{(1.25)}$ | $60.97_{(1.02)}$ | $55.39_{(1.83)}$ | $66.24_{(2.40)}$ | $60.78_{(2.91)}$ | $65.72_{(1.19)}$ | $62.38_{(1.67)}$ |
| ProtoNet | $72.99_{(0.56)}$ | $68.22_{(0.46)}$ | $61.34_{(1.08)}$ | $57.41_{(0.76)}$ | $68.92_{(1.64)}$ | $64.81_{(1.95)}$ | $65.29_{(0.82)}$ | $63.73_{(1.18)}$ |
| MAML | $79.59_{(0.33)}$ | $75.63_{(0.18)}$ | $70.49_{(0.54)}$ | $68.63_{(1.51)}$ | $68.38_{(1.27)}$ | $65.82_{(2.49)}$ | $68.43_{(1.85)}$ | $66.75_{(1.62)}$ |
| Siamese | $80.40_{(0.29)}$ | $65.00_{(11.69)}$ | $71.10_{(1.68)}$ | $51.43_{(2.83)}$ | $59.96_{(3.56)}$ | $50.00_{(0.19)}$ | - | - |
| EGNN | $80.11_{(0.31)}$ | $75.71_{(0.21)}$ | $71.24_{(0.37)}$ | $66.36_{(0.29)}$ | $68.84_{(1.35)}$ | $62.72_{(1.97)}$ | $66.42_{(0.77)}$ | $63.98_{(1.20)}$ |
| IterRefLSTM | $81.10_{(0.10)}$ | $80.97_{(0.06)}$ | $69.63_{(0.16)}$ | $71.73_{(0.06)}$ | $49.56_{(2.32)}$ | $48.54_{(1.48)}$ | - | - |
| PAR | $82.13_{(0.26)}$ | $\underline{80.02}_{(0.30)}$ | $\underline{75.15}_{(0.35)}$ | $\underline{72.33}_{(0.47)}$ | $68.08_{(2.23)}$ | $65.62_{(3.49)}$ | $70.01_{(0.85)}$ | $\underline{68.22}_{(1.34)}$ |
| ADKF-IFT | $\underline{82.43}_{(0.60)}$ | $77.94_{(0.91)}$ | $67.72_{(1.21)}$ | $58.69_{(1.44)}$ | $\mathbf{98.18}_{(3.05)}$ | $\underline{67.04}_{(4.86)}$ | $\underline{72.07}_{(0.81)}$ | $67.50_{(1.23)}$ |
| PACIA | $\mathbf{84.25}_{(0.31)}$ | $\mathbf{82.77}_{(0.15)}$ | $\mathbf{82.40}_{(0.26)}$ | $\mathbf{77.72}_{(0.34)}$ | $\underline{72.58}_{(2.23)}$ | $\mathbf{68.80}_{(4.01)}$ | $\mathbf{72.38}_{(0.96)}$ | $\mathbf{69.89}_{(1.17)}$ |

## 5.2 PERFORMANCE COMPARISON ON FS-MOL

**Setting.** We also perform experiments on FS-Mol (Stanley et al., 2021), a new benchmark consisting of a large number of diverse tasks for model pretraining and a set of few-shot tasks with imbalanced classes. We adopt the public data split provided by (Stanley et al., 2021). Each support set contains 64 labeled molecules, and can be imbalanced where the number of labeled molecules from inactive and active may not be equal. All remaining molecules in the task form the query set. Following Schimunek et al. (2023), testing tasks are divided into categories with support size 16, which is close to real-world scenario. The performance is evaluated by ΔAUPRC (change in area under the precision-recall curve), averaged across meta-testing tasks.

**Baselines.** The set of baselines is the same with MoleculeNet's.

Table 2: Test ΔAUPRC obtained on FS-Mol. Tasks are categorized by target protein type and the number of tasks per category is reported in brackets. The best results are bolded, second-best results are underlined.

| Method | All [157] | Kinases [125] | Hydrolases [20] | Oxidoreductases [7] |
|---|---|---|---|---|
| Random Forest | $.092_{(.007)}$ | $.081_{(.009)}$ | $.158_{(.028)}$ | $.080_{(.029)}$ |
| GNN-ST | $.029_{(.004)}$ | $.027_{(.004)}$ | $.040_{(.018)}$ | $.020_{(.016)}$ |
| MAT | $.052_{(.005)}$ | $.043_{(.005)}$ | $.095_{(.019)}$ | $.062_{(.024)}$ |
| GNN-MT | $.093_{(.006)}$ | $.093_{(.006)}$ | $.108_{(.025)}$ | $.053_{(.018)}$ |
| MAML | $.159_{(.009)}$ | $.177_{(.009)}$ | $.105_{(.024)}$ | $.054_{(.028)}$ |
| PAR | $.164_{(.008)}$ | $.182_{(.009)}$ | $.109_{(.020)}$ | $.039_{(.008)}$ |
| ProtoNet | $.207_{(.008)}$ | $.215_{(.009)}$ | $.209_{(.030)}$ | $.095_{(.029)}$ |
| EGNN | $.212_{(.011)}$ | $.224_{(.010)}$ | $.205_{(.024)}$ | $.097_{(.022)}$ |
| Siamese | $.223_{(.010)}$ | $.241_{(.010)}$ | $.178_{(.026)}$ | $.082_{(.025)}$ |
| IterRefLSTM | $\underline{.234}_{(.010)}$ | $\mathbf{.251}_{(.010)}$ | $.199_{(.026)}$ | $.098_{(.027)}$ |
| ADKF-IFT | $\underline{.234}_{(.009)}$ | $.248_{(.020)}$ | $\underline{.217}_{(.017)}$ | $\mathbf{.106}_{(.008)}$ |
| PACIA | $\mathbf{.236}_{(.008)}$ | $\mathbf{.251}_{(.016)}$ | $\mathbf{.219}_{(.029)}$ | $\mathbf{.106}_{(.010)}$ |

**Performance.** Table 2 shows the results. We find that PACIA performs the best while ADKF-IFT obtains comparable performance. However, the time-efficiency of PACIA is much higher since the adaptation only needs a single forward pass. While ADKF-IFT takes multiple local-update steps (about 5 times faster in both meta-training and testing, with default numbers of local-update steps).

## 5.3    ABLATION STUDY

We consider various variants of PACIA, including (i) **fine-tuning**: use the same model structure and fine-tuning all parameters to adapt to each property without hypernetworks; (ii) **w/o P**: removing property-level adaptation, thus the GNN encoder will not be adapted by the hypernetwork for each property; and (iii) **w/o M**: removing molecule-level adaptation, that all molecules are processed by the same predictor.

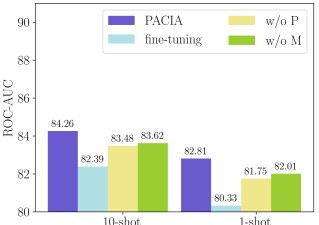
(a) Different adaptation strategies.

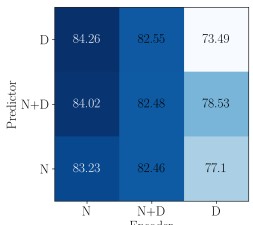
(b) Different modulation functions.

Figure 2: Ablation study on 10-shot tasks of Tox21.

Figure 2(a) provides performance comparison on Tox21. Observations are as follows: (i) The performance gain of PACIA over "w/o M" shows the necessity of molecule-level adaptation; (ii) The gap between PACIA and "w/o P" indicates the effect of adapting the model to be property-specific; (iii) One can also notice that without molecule-level adaptation, "w/o M" still obtains better performance than gradient-based baselines like PAR, which indicates the advantage of designing the amortization-based hypernetwork; and (iv) The poor performance of "fine-tuning" is possibly because of the overfitting caused by updating all parameters with only a few samples. In summary, every component of PACIA is important to obtaining good performance.

Now that the effectiveness of property-level and molecule-level adaptation are validated, we further investigate modulation functions, i.e, modulating node embedding (**N**), modulating propagation depth (**D**), modulating both (**ND**), for encoder and predictor. There are $3 \times 3$ combinations whose performance is presented in Figure 2(b). We find that only modulating node embedding in encoder while only modulating propagation depth in predictor obtains the best performance. We interpret this result as the GIN encoder has highly non-linearity across layers, where truncation would lead to non-explainability and somehow perturb the black-box. While the operation of relation graph in predictor updates node embedding in a linear way (5), adapting the propagation depth is harmonious with its message passing function.

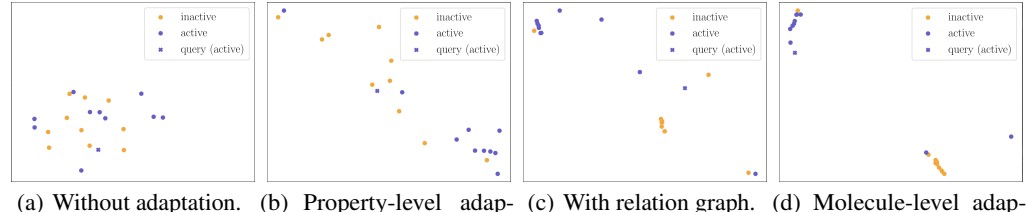
(a) Without adaptation.    (b) Property-level adaptated.    (c) With relation graph.    (d) Molecule-level adaptated.

Figure 3: Molecular representation visualization for 10-shot case in task SR-p53 of Tox21. Pretrained GIN is used as encoder.

Figure 3 shows the t-SNE visualization (Van der Maaten & Hinton, 2008) of molecular representations learned on a 10-shot support set and a query molecule with ground truth label "active" in task SR-p53 from Tox21. As shown, molecular representations obtained using pre-trained encoder without adaptation (Figure 3(a)) are mixed up, since the encoder has not been adapted to the target property of the task. Molecular representations being processed by our property-adaptive GNN encoder (Figure 3(b)) becomes more distinguishable, indicating that adapting molecular representation in property-level takes effect. Molecular representations in Figure 3(c) and Figure 3(d) form clear clusters as we encourage similar molecules to be connected during relation graph refinement by (5). The difference is that molecular representations in Figure 3(c) are refined by the best depth number for all tasks in 10-shot case, while molecular representations in Figure 3(d) are refined by 4 layers which are selected for the specific query molecule. As shown, we can conclude that our molecular-adaptive refinement layers help better separate molecules of different classes.

We also have ablation study of configurations inside hypernetwork, provided in Appendix C.4.

### 5.4 A CLOSER LOOK AT HIERARCHICAL ADAPTATION MECHANISM

**Property-level Adaptation.** Our parameter-efficient property-level adaptation is achieved by using hypernetworks to modulate the node embeddings during message passing. We compared this amortization-based adaptation with gradient-based adaptation in PAR which has similar main network with PACIA. The results are shown in Table 3. We record their adaptation process, i.e., time required to process the support set and the test performance. PAR uses the molecules in support set to take gradient steps, and updates all parameters in GNN. We record each of a maximum five steps, where we can find that it overfits easily as the performance keeps dropping with more steps. The time consumption also grows. In contrast, PACIA processes molecules in support set by hypernetwork, which is much more efficient as only one single forward pass is needed. PACIA can obtain better performance due to the reduction of adaptive parameters, which also leads to better generalization and alleviates the risk of overfitting to the few shots. Table 3 and Figure 1(a) both indicate that the underlying overfitting problem which can be mitigated by PACIA.

Table 3: Comparison of property-level adaptation approaches.

|  | PACIA | PAR | | | | |
|---|---|---|---|---|---|---|
| Total parameters | 3.28M | 2.31M | | | | |
| Adaptive parameters | 3.00K | 0.38M | | | | |
| Adaptation steps | - | 1 | 2 | 3 | 4 | 5 |
| Testing ROC-AUC | 84.26 | 82.07 | 81.85 | 80.32 | 79.09 | 77.25 |
| Time (secs) | 1.09 | 2.02 | 3.62 | 5.34 | 6.76 | 8.10 |

**Molecule-level Adaptation.** Here, we present a case study on molecule-level adaptation. More experiments on validating the design of molecule-level adaptation is in Appendix C.5. We use a 1-shot support set and 3 query molecules in task SR-p53 of Tox21. In Figure 4 (a), $x_1$ and $x_0$ are support molecules with different labels, $q_1$, $q_2$ and $q_3$ are query molecules. As shown, classifying $q_1$ and $q_3$ is relatively easy and the adapted propagation depth will be 1, while classifying $q_2$ is hard and requires 4 layers to propagate. Considering the shared substructures (function groups), $q_1$ and $x_1$ are visually similar, $q_3$ and $x_0$ are visually similar. While both $x_1$ and $x_0$ share substructures with $q_2$, it is hard to tell which of them is more similar to $q_2$. Figure 4 (a) provides the cosine similarity based on the molecule representations generated by Pre-GNN , which confirms our observation: $q_1$ is much more similar to $x_1$, $q_3$ is much more similar to $x_0$, and $q_2$ has close similarities with the both samples. Intuitively, classifying $q_1$ and $q_3$ will be easier while $q_2$ will be hard. In the dynamic propagation of PACIA, we find different layers are taken: $t' = 1$ for both $q_1$ and $q_3$ while $t' = 4$ for $q_2$. PACIA achieves effective molecule-level adaptation by assigning more complex models for molecules that are difficult to classify.

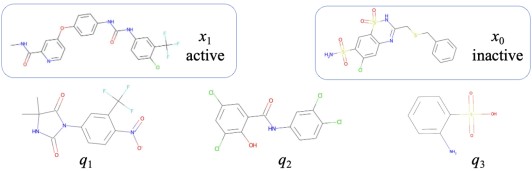

| Molecule | $q_1$ | | $q_2$ | | $q_3$ | |
|---|---|---|---|---|---|---|
|  | $x_1$ | $x_0$ | $x_1$ | $x_0$ | $x_1$ | $x_0$ |
| Cosine similarity | 0.183 | -0.024 | 0.278 | 0.252 | -0.031 | 0.129 |
| $l'$ | 1 | | 4 | | 1 | |

Figure 4: Illustration of molecule-level adaptation. (a), Molecular graphs of support molecules $x_1$, $x_0$ and query molecules $q_1$, $q_2$, $q_3$. (b), Cosine similarities between query molecules and support molecules, and propagation depth taken to classify each query molecule.

## 6 CONCLUSION

We propose PACIA to handle few-shot MPP in a parameter-efficient manner. We investigate two key factors in few-shot molecular property prediction with the common encoder-predictor framework: adaptation-efficiency and molecular-level adaptation. Evidence shows that too much adaptive parameter would lead to overfitting, thus we design a parameter-efficient GNN adapter, which can modulate node embedding and propagation depth of message passing of GNN in a unified way. We also notice the importance of capturing molecule-level difference and therefore propose hierarchical adaptation mechanism, which is achieved by using unified GNN adapter in both encoder and predictor. Empirical results show that PACIA achieves the best performance on both MoleculeNet and FS-Mol.

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

## A MORE DETAILS OF PACIA

### A.1 ENCODER

**Encoder for MoleculeNet.** As the main network of the encoder, to process a molecule with GNN, each node embedding $\mathbf{h}_v$ represents an atom, and each edge $e_{vu}$ represents a chemical bond. Here, we use GIN (Xu et al., 2019) as the main network in encoder, which is a powerful GNN structure. In GIN, the aggregation function in (1) is specified as adding all neighbors up: and for the update function is adding the aggregated embeddings and the target node, and feeding to a MLP:

$$\mathbf{h}_v^l = \text{MLP}_G^l \left( (1 + \epsilon)\mathbf{h}_v^{l-1} + \sum_{u \in \mathcal{H}(v)} \mathbf{h}_u^{l-1} \right), \tag{14}$$

where $\epsilon$ is a scalar parameter to distinguish the target node. To obtain the molecular representation, the readout function in (2) is specified as

$$\mathbf{r} = \text{MLP}_R\left(\text{MEAN}(\{\mathbf{h}_v^L | v \in \mathcal{V}\})\right). \tag{15}$$

**Encoder for FS-Mol.** Following existing works (Chen et al., 2022; Schimunek et al., 2023), we directly adopt the PNA (Corso et al., 2020) network provided in FS-Mol benchmark (Stanley et al., 2021) as the molecular encoder.

## A.2 PREDICTOR

The classifier needs to make prediction of the query $\hat{\mathbf{h}}_{\tau,q}$, according to the $N_\tau$ labeled support samples $\{(\mathbf{h}_{\tau,s}, y_{\tau,s}) \mid \mathcal{X}_{\tau,s} \in \mathcal{S}_\tau\}$. We adopt an adaptive classifier (Requeima et al., 2019), which map the labeled samples in each class to the parameters of a linear classifier, i.e.,

$$\boldsymbol{w}_\pm = \text{MLP}_w\left(\frac{1}{|\mathcal{S}_\tau^\pm|}\sum\nolimits_{\mathcal{X}_{\tau,s} \in \mathcal{S}_\tau^\pm} \mathbf{h}_{\tau,s}\right), \; b_\pm = \text{MLP}_b\left(\frac{1}{|\mathcal{S}_\tau^\pm|}\sum\nolimits_{\mathcal{X}_{\tau,s} \in \mathcal{S}_\tau^\pm} \mathbf{h}_{\tau,s}\right), \tag{16}$$

where $\boldsymbol{w}_\pm$ has the same dimension with $\mathbf{h}_{\tau,q}$, and $b_\pm$ is a scalar. Then the prediction is made by

$$\hat{\boldsymbol{y}}_{\tau,q} = \text{softmax}([\boldsymbol{w}_-^\top \mathbf{h}_{\tau,q} + b_-, \; \boldsymbol{w}_+^\top \mathbf{h}_{\tau,q} + b_+]), \tag{17}$$

where $\text{softmax}(\boldsymbol{x}) = \exp(\mathbf{x})/\sum_i \exp([\boldsymbol{x}]_i)$ and $[\boldsymbol{x}]_i$ means the $i$th element in $\boldsymbol{x}$.

## A.3 UNIFIED GNN ADAPTER

The choice of $e$ can be various (Wu et al., 2023), while in this work we adopt a simple feature-wise linear modulation (FiLM) (Perez et al., 2018) function.

## A.4 HYPERPARAMETERS

Here we provide the detailed hyperparameter setting of PACIA.

**Hyperparameters on MoleculeNet.** The maximum layer number of the GNN $L_{\text{enc}} = 5$, the maximum depth of the relation graph $L_{\text{rel}} = 5$, During training, for each layer in GNN, we set dropout rate as 0.5 operated between the graph operation and FiLM layer. The dropout rate of MLP in (9) (10) and (11) is 0.1. For all baselines, we use Adam optimizer (Kingma & Ba, 2015) with learning rate 0.006 and the maximum episode number is 25000. In each episode, the meta-training tasks are learned one-by-one, query set size $M = 16$. The ROC-AUC is evaluated every 10 epochs on meta-testing tasks and the best performance is reported. Table 4 shows the details of the other parts. Experiments are conducted on a 24GB NVIDIA GeForce RTX 3090 GPU, with Python 3.8.13, CUDA version 11.7, Torch version 1.10.1.

**Hyperparameters on FS-Mol.** The maximum layer number of the GNN $L_{\text{enc}} = 8$, the maximum depth of the relation graph $L_{\text{rel}} = 5$, During training, the dropout rate of MLP$_L$ is 0.1. We use Adam optimizer (Kingma & Ba, 2015) with learning rate 0.0001 and the maximum episode number is 3000. In each episode, the meta-training tasks are learned with batch size 16, support set size $N_\tau = 64$, and the others are used as queries. The average precision s evaluated every 50 epochs on validation tasks and the the model with best validation performance is tested and reported. Table 5 shows the details of the other parts.

## B ADOPTING MAML FOR PROPERTY-LEVEL ADAPTATION

Denote all model parameter as $\boldsymbol{\Theta}$. The model first predict samples in support set and get loss to do local-update. Denote the loss for local-update as $\mathcal{L}_\tau^S(\boldsymbol{\Theta}) = \sum_{\mathcal{X}_{\tau,s} \in \mathcal{S}_\tau} \boldsymbol{y}_{\tau,s}^\top \log(\hat{\boldsymbol{y}}_{\tau,s})$, where $\hat{\boldsymbol{y}}_{\tau,s}$ is the prediction made by the main network with parameter $\boldsymbol{\Theta}$. The loss for global-update is calculated with samples in query set, denoted as , $\mathcal{L}_\tau^Q(\boldsymbol{\Theta}_\tau') = \sum_{\mathcal{X}_{\tau,q} \in \mathcal{Q}_\tau} \boldsymbol{y}_{\tau,q}^\top \log(\hat{\boldsymbol{y}}_{\tau,q})$, where $\hat{\boldsymbol{y}}_{\tau,q}$ is the prediction made by the main network with parameter $\boldsymbol{\Theta}_\tau'$. Then Algorithm 3 can be adopted for meta-training, and Algorithm 4 for meta-testing.

Table 4: Details of model structure for MoleculeNet.

| | Layers | Output Dimension |
|---|---|---|
| MLP in (9) | input $\frac{1}{|\mathcal{V}_{\tau,s}|} \sum_{v \in \mathcal{X}_{\tau,s}} [\mathbf{h}_v^l \mid \boldsymbol{y}_{\tau,s}]$, fully connected, LeakyReLU | 300 |
| | 2×fully connected with with residual skip connection, $\frac{1}{K}[\sum_{\mathcal{X}_{\tau,s} \in \mathcal{S}_\tau^+}(\cdot) \mid \sum_{\mathcal{X}_{\tau,s} \in \mathcal{S}_\tau^-}(\cdot)]$ | 300 |
| MLP in (10) | 3×fully connected with residual skip connection | 601 |
| MLP in (11) | 3×fully connected with residual skip connection | 257 |
| $\text{MLP}_G^l$ in (14) | input $(1+\epsilon)\mathbf{h}_v^{l-1} + \mathbf{h}_{\text{agg}}^{l-1}$, fully connected, ReLU | 600 |
| | fully connected | 300 |
| $\text{MLP}_L$ in (15) | input READOUT $(\{\mathbf{h}_v^T \mid v \in \mathcal{V}_{t,i}\})$, fully connected, LeakyReLU | 128 |
| | fully connected | 128 |
| MLP in (4) | input $\exp(|\mathbf{h}_{\tau,i}^{l-1} - \mathbf{h}_{\tau,i}^{l-1}|)$, fully connected, LeakyReLU | 256 |
| | fully connected, LeakyReLU | 128 |
| | fully connected | 1 |
| MLP in (5) | fully connected, LeakyReLU | 256 |
| | fully connected, LeakyReLU | 128 |
| $\text{MLP}_w$ in (16) | input $\frac{1}{K} \sum_{y_{\tau,s}=c} \boldsymbol{h}_{\tau,s}$, fully connected with residual skip connection, LeakyReLU | 128 |
| | 2× (fully connected with residual skip connection, LeakyReLU) | 128 |
| | fully connected | 128 |
| $\text{MLP}_b$ in (16) | input $\frac{1}{K} \sum_{y_{\tau,s}=c} \boldsymbol{h}_{\tau,s}$, fully connected with residual skip connection, LeakyReLU | 128 |
| | 2× (fully connected with residual skip connection, LeakyReLU) | 128 |
| | fully connected | 1 |

Table 5: Details of model structure for FS-Mol.

| | Layers | Output Dimension |
|---|---|---|
| MLP in (9) | input $\frac{1}{|\mathcal{V}_{\tau,s}|} \sum_{v \in \mathcal{X}_{\tau,s}} [\mathbf{h}_v^l \mid \boldsymbol{y}_{\tau,s}]$,, fully connected, LeakyReLU | 512 |
| | 2× fully connected with with residual skip connection, $[\frac{1}{|\mathcal{S}_\tau^+|} \sum_{\mathcal{X}_{\tau,s} \in \mathcal{S}_\tau^+}(\cdot) \mid \frac{1}{|\mathcal{S}_\tau^-|} \sum_{\mathcal{X}_{\tau,s} \in \mathcal{S}_\tau^-}(\cdot)]$ | 512 |
| MLP in (10) | 3×fully connected with residual skip connection | 1025 |
| MLP in (11) | 3×fully connected with residual skip connection | 513 |
| MLP in (4) | input $\exp(|\mathbf{h}_{\tau,i}^{l-1} - \mathbf{h}_{\tau,i}^{l-1}|)$, fully connected, LeakyReLU | 256 |
| | fully connected, LeakyReLU | 128 |
| | fully connected | 1 |
| MLP in (5) | fully connected, LeakyReLU | 256 |
| | fully connected, LeakyReLU | 256 |
| $\text{MLP}_w$ in (16) | input $\frac{1}{|\mathcal{S}_\tau^\pm|} \sum_{y_{\tau,s}=c} \boldsymbol{h}_{\tau,s}$, fully connected with residual skip connection, LeakyReLU | 256 |
| | 2× (fully connected with residual skip connection, LeakyReLU) | 256 |
| | fully connected | 256 |
| $\text{MLP}_b$ in (16) | input $\frac{1}{|\mathcal{S}_\tau^\pm|} \sum_{y_{\tau,s}=c} \boldsymbol{h}_{\tau,s}$, fully connected with residual skip connection, LeakyReLU | 256 |
| | 2× (fully connected with residual skip connection, LeakyReLU) | 256 |
| | fully connected | 1 |

---

**Algorithm 3** Meta-training with MAML

**Input:** meta-training task set $\mathcal{T}_{\text{train}}$
1: initialize $\Theta$ randomly;
2: **while** not done **do**
3:     **for** each task $\mathcal{T}_\tau \in \mathcal{T}_{\text{train}}$ **do**
4:         evaluate $\nabla_\Theta \mathcal{L}_\tau^S(\Theta)$ with respect to all samples in $\mathcal{S}_\tau$;
5:         compute adapted parameters with gradient descent: $\Theta_\tau' = \Theta - \nabla_\Theta \mathcal{L}_\tau^S(\Theta)$;
6:     **end for**
7:     update $\Theta \leftarrow \Theta - \nabla_\Theta \sum_{\mathcal{T}_\tau \in \mathcal{T}_{\text{train}}} \mathcal{L}_\tau^Q(\Theta_\tau')$;
8: **end while**
9: **return** learned $\Theta^*$.

---

**Algorithm 4** Meta-testing with MAML

---

**Input:** learned $\boldsymbol{\Theta}*$, a meta-testing task $\mathcal{T}_\tau$;
1: evaluate $\nabla_{\boldsymbol{\Theta}} \mathcal{L}_\tau^S(\boldsymbol{\Theta})$ with respect to all samples in $\mathcal{S}_\tau$;
2: compute adapted parameters with gradient descent: $\boldsymbol{\Theta}'_\tau = \boldsymbol{\Theta} - \nabla_{\boldsymbol{\Theta}} \mathcal{L}_\tau^S(\boldsymbol{\Theta})$;
3: make prediction $\boldsymbol{y}_{\tau,q}$ for $\mathcal{X}_{\tau,q} \in \mathcal{Q}_\tau$ with adapted parameter $\boldsymbol{\Theta}'_\tau$;

---

### B.1 Comparison with Existing Works

We compare the proposed PACIA with existing few-shot MPP approaches in Table 6. As shown, we manage to compare in perspectives of support of pre-training, property-level adaptation, molecule-level adaptation. fast-adaptation and adaptation strategy. With the help of hypernetworks, our method not only introduces novel molecule-level adaptation, but also can adapt on property-level more effectively and efficiently.

Table 6: Comparison of the proposed PACIA with existing few-shot MPP methods.

| Method | Support Pre-training | Hierarchical adaptation Property-level | molecule-level | Fast adaptation | Adaptation Strategy |
|---|---|---|---|---|---|
| IterRefLSTM | × | ✓ | × | ✓ | Pair-wise similarity |
| Meta-MGNN | ✓ | ✓ | × | × | Gradient |
| PAR | ✓ | ✓ | × | × | Attention+Gradient |
| ADKF-IFT | ✓ | ✓ | × | × | Gradient+statistical learning |
| MHNfs | ✓ | ✓ | ✓ | ✓ | Attention+pair-wise similarity |
| GS-META | ✓ | × | ✓ | ✓ | Message passing |
| PACIA | ✓ | ✓ | ✓ | ✓ | Hypernetwork |

From the perspective of hypernetwork, the usage of the hypernetwork for encoder is related to GNN-FiLM (Brockschmidt, 2020), which considers a GNN as main network. It builds hypernetwork with target node as input to generate parameters of FiLM layers, to equip different nodes with different aggregation functions in the GNN. What and how to adapt are similar to ours, but it is different that the input of our hypernetwork for encoder is $\mathcal{S}_\tau$ and how we encode a set of labeled graphs.

In few-shot learning, some recent works (Requeima et al., 2019; Lin et al., 2021) use hypernetworks to process the task context to make the model task-adaptive. Their hypernetworks have similar functionality of our hypernetwork for encoder, that are used to map the support set to parameter to modulate the main network, but their main networks are convolutional neural network (CNN) and MLP respectively, there is significant difference about what to modulate. As for the usage of the hypernetwork for predictor, which is used to evaluate an unlabeled sample with a set of labeled ones to encode model architecture, did not appear in the literature.

## C More Details of Experiments

### C.1 Datasets

**MoleculeNet.** There are four sub-datasets for few-shot MPP: Tox21 (National Center for Advancing Translational Sciences, 2017), SIDER (Kuhn et al., 2016), MUV (Rohrer & Baumann, 2009) and ToxCast (Richard et al., 2016), which are included in MoleculeNet (Wu et al., 2018). We adopt the task splits provided by existing works(Altae-Tran et al., 2017; Wang et al., 2021). Tox21 is a collection of nuclear receptor assays related to human toxicity, containing 8014 compounds in 12 tasks, among which 9 are split for training and 3 are split for testing. SIDER collects information about side effects of marketed medicines, and it contains 1427 compounds in 21 tasks, among which 21 are split for training and 6 are split for testing. MUV contains compounds designed to be challenging for virtual screening for 17 assays, containing 93127 compounds in 17 tasks, among which 12 are split for training and 5 are split for testing. ToxCast collects compounds with toxicity labels, containing 8615 compounds in 617 tasks, among which 450 are split for training and 167 are split for testing.

**FS-Mol.** FS-Mol benchmark, which contains a set of few-shot learning tasks for molecular property prediction carefully collected from ChEMBL27 (Mendez et al., 2019) by Stanley et al. (2021). Following existing works (Chen et al., 2022; Schimunek et al., 2023), we use the same 10% of all tasks which contains 233,786 unique compounds, split into training (4,938 tasks), validation (40 tasks), and test (157 tasks) sets. Each task is associated with a protein target.

## C.2 Baselines

We compare our method with following baselines:

- **Siamese** (Koch et al., 2015): It learns two neural networks which are symmetric on structure to identity whether the input molecule pairs are from the same class. The performance is copied from (Altae-Tran et al., 2017) due to the lack of code.

- **ProtoNet**[1] (Snell et al., 2017): It makes classification according to inner-product similarity between the target and the prototype of each class. This method is incorporated as a classifier after the GNN encoder.

- **MAML**[2] (Finn et al., 2017): It learns a parameter initialization and the model is adapted to each task via few gradient steps on the support set. We adopt this method for all parameters in a model composed of a GNN encoder and a linear classifier.

- **EGNN**[3] (Kim et al., 2019): It builds a relation graph that samples are refined, and it learns to predict edge-labels in the relation graph. This method is incorporated as the predictor after the GNN encoder.

- **GNN-FiLM**(Brockschmidt, 2020): GNN-FiLM has built hypernetwork with target node as input to generate parameters of FiLM layers, which equips different nodes with different aggregation functions in the GNN. We adopt this as encoder and a MLP as classifier and train on all samples in meta-training tasks and samples in support set of all meta-testing tasks;

- **IterRefLSTM** (Altae-Tran et al., 2017): It introduces matching networks combined with long short-term memory (LSTM) to refine the molecular representations according to the task context. The performance is copied from (Altae-Tran et al., 2017) due to the lack of code.

- **PAR**[4] (Wang et al., 2021): It introduces an attention mechanism to capture task-dependent property and an inductive relation graph between samples, and incorporates MAML to train.

- **ADKF-IFT**[5] (Wang et al., 2021): It adopt gradient-based strategy to learn the encoder where it proposes Implicit Function Theory to avoid computing the hyper-gradient. And a Gaussian Process is learned from scratch in each task as classifier.

- **Pre-GNN**[6] (Hu et al., 2019): It trains a GNN encoder on ZINC15 dataset with graph-level and node-level self-supervised tasks, and fine-tunes the pre-trained GNN on downstream tasks. We adopt the pre-trained GNN encoder and a linear classifier.

- **GraphLoG**[7] (Xu et al., 2021): It introduces hierarchical prototypes to capture the global semantic clusters. And adopts an online expectation-maximization algorithm to learn. We adopt the pre-trained GNN encoder and a linear classifier.

- **MGSSL**[8] (Hu et al., 2019): It trains a GNN encoder on ZINC15 dataset with graph-level, node-level and motif-level self-supervised tasks, and fine-tunes the pre-trained GNN on downstream tasks. We adopt the pre-trained GNN encoder and a linear classifier.

- **GraphMAE**[9] (Hou et al., 2022): It presents a masked graph autoencoder for generative self-supervised graph pre-training and focus on feature reconstruction with both a masking strategy and scaled cosine error. We adopt the pre-trained GNN encoder and a linear classifier.

---

[1] https://github.com/jakesnell/prototypical-networks
[2] https://github.com/learnables/learn2learn
[3] https://github.com/khy0809/fewshot-egnn
[4] https://github.com/tata1661/PAR-NeurIPS21
[5] https://github.com/Wenlin-Chen/ADKF-IFT
[6] http://snap.stanford.edu/gnn-pretrain
[7] http://proceedings.mlr.press/v139/xu21g/xu21g-supp.zip
[8] https://github.com/zaixizhang/MGSSL
[9] https://github.com/THUDM/GraphMAE

- **Meta-MGNN**[10] (Guo et al., 2021): It incorporates self-supervised tasks such as bond reconstruction and atom type prediction to be jointly optimized via MAML. It uses the pre-trained GNN encoder provided by (Hu et al., 2019).

- **Pre-PAR**: The same as PAR but uses the pre-trained GNN encoder provided by (Hu et al., 2019).

- **Pre-ADKF-IFT**: The same as ADKF-IFT but uses the pre-trained GNN encoder provided by (Hu et al., 2019).

## C.3 PERFORMANCE COMPARISON WITH PRE-TRAINING

**Baselines with Pre-training.** We compare with the following baselines with (w/) pre-training: (i) Methods which fine-tune pre-train GNN encoders, including **Pre-GNN** (Hu et al., 2019), **GraphLoG** (Xu et al., 2021), **MGSSL** (Zhang et al., 2021), **GraphMAE** (Hou et al., 2022); (ii) Few-shot MPP methods incorporating pre-trained encoders provided by (Hu et al., 2019), including **Meta-MGNN** (Guo et al., 2021), **Pre-PAR** (Wang et al., 2021) and **Pre-ADKF-IFT** Chen et al. (2022). All encoders have the same structure (Hu et al., 2019) and are pre-trained on ZINC15 dataset (Sterling & Irwin, 2015). We equip our PACIA with the same pre-trained encoder, and name it as **Pre-PACIA**.

Table 7: Test ROC-AUC obtained with pre-trained GNN encoder.

| Method | Tox21 | | SIDER | | MUV | | ToxCast | |
|---|---|---|---|---|---|---|---|---|
| | 10-shot | 1-shot | 10-shot | 1-shot | 10-shot | 1-shot | 10-shot | 1-shot |
| Pre-GNN | $83.02_{(0.13)}$ | $82.75_{(0.09)}$ | $77.55_{(0.14)}$ | $67.34_{(0.30)}$ | $67.22_{(2.16)}$ | $65.79_{(1.68)}$ | $73.03_{(0.67)}$ | $71.26_{(0.85)}$ |
| GraphLoG | $81.61_{(0.35)}$ | $79.23_{(0.93)}$ | $75.18_{(0.27)}$ | $67.52_{(1.40)}$ | $67.83_{(1.65)}$ | $66.56_{(1.46)}$ | $73.92_{(0.15)}$ | $73.10_{(0.39)}$ |
| MGSSL | $83.24_{(0.09)}$ | $\underline{83.21}_{(0.12)}$ | $77.87_{(0.18)}$ | $69.66_{(0.21)}$ | $68.58_{(1.32)}$ | $66.93_{(1.74)}$ | $73.51_{(0.45)}$ | $72.89_{(0.63)}$ |
| GraphMAE | $84.01_{(0.27)}$ | $81.54_{(0.18)}$ | $76.07_{(0.15)}$ | $67.60_{(0.38)}$ | $67.99_{(1.28)}$ | $\underline{67.50}_{(2.12)}$ | $74.15_{(0.33)}$ | $72.67_{(0.71)}$ |
| Meta-MGNN | $83.44_{(0.14)}$ | $82.67_{(0.20)}$ | $77.84_{(0.34)}$ | $74.62_{(0.41)}$ | $68.31_{(3.06)}$ | $66.10_{(3.98)}$ | $74.69_{(0.57)}$ | $73.29_{(0.85)}$ |
| Pre-PAR | $84.95_{(0.24)}$ | $\underline{83.01}_{(0.28)}$ | $\underline{78.05}_{(0.15)}$ | $\underline{75.29}_{(0.32)}$ | $69.88_{(1.57)}$ | $66.96_{(2.63)}$ | $75.48_{(0.99)}$ | $\underline{73.90}_{(1.21)}$ |
| Pre-ADKF-IFT | $\underline{86.06}_{(0.35)}$ | $80.97_{(0.48)}$ | $70.95_{(0.60)}$ | $62.16_{(1.03)}$ | $\mathbf{95.74}_{(0.37)}$ | $67.25_{(3.87)}$ | $\mathbf{76.22}_{(0.13)}$ | $71.13_{(1.15)}$ |
| Pre-PACIA | $\mathbf{86.40}_{(0.27)}$ | $\mathbf{84.35}_{(0.14)}$ | $\mathbf{83.97}_{(0.22)}$ | $\mathbf{80.70}_{(0.28)}$ | $\underline{73.43}_{(1.96)}$ | $\mathbf{69.26}_{(2.35)}$ | $\mathbf{76.22}_{(0.73)}$ | $\mathbf{75.09}_{(0.95)}$ |

**Performance with Pre-training.** Table 7 shows the results. We can see that Pre-PACIA obtains significantly better performance except the 10-shot case on MUV, surpassing the second-best method Pre-ADKF-IFT by 3.10%. MGSSL defeats the other methods which fine-tune pre-trained GNN encoders, i.e., Pre-GNN, GraphLoG, and GraphMAE. However, it still performs worse than Pre-PACIA equipped with Pre-GNN, which validates the necessity of designing a few-shot MPP method instead of simply fine-tuning a pre-trained GNN encoder. Moreover, comparing Pre-PACIA and PACIA in Table 1, the pre-trained encoder brings 3.05% improvement in average performance due to a better starting point of learning.

## C.4 ABLATION STUDY OF HYPERNETWORK

For the specific settings, please refer to Table 4 and Table 5 for hyperparameters chosen on MoleculeNet and FS-Mol, where the configuration of hypernetwork is included.

For ablation studies, hereresults concerning with three aspects in hypernetwork:

## C.4.1 EFFECT OF CONCATENATING LABEL

We show effect of concatenating label $\boldsymbol{y}_{\tau,s}$. Table 8 shows the testing ROC-AUC obtained on SIDER. As shown, "w/ Label" helps keeping the label information in support set, which improves the performance.

---

[10] https://github.com/zhichunguo/Meta-Meta-MGNN

Table 8: Effect of concatenating label $\boldsymbol{y}_{\tau,s}$, testing ROC-AUC obtained on SIDER.

|  | 10-shot | 1-shot |
|---|---|---|
| w/ Label | $82.40_{(0.26)}$ | $77.72_{(0.34)}$ |
| w/o Label | $76.91_{(0.17)}$ | $74.10_{(0.41)}$ |

### C.4.2  DIFFERENT WAYS OF COMBINING PROTOTYPES

We show performance with different ways of combining active prototype $\boldsymbol{r}_{\tau,+}^l$ and inactive prototype $\boldsymbol{r}_{\tau,-}^l$ in Equation (10)—(11). Table 9 show the results. As "Concatenating" active and inactive prototypes allows MLP to capture more complex patterns, it obtains better performance on SIDER as shown in Table 9.

Table 9: Different ways of combining active prototype $\boldsymbol{r}_{\tau,+}^l$ and inactive prototype $\boldsymbol{r}_{\tau,-}^l$, testing ROC-AUC obtained on SIDER.

|  | 10-shot | 1-shot |
|---|---|---|
| Concatenating | $82.40_{(0.26)}$ | $77.72_{(0.34)}$ |
| Mean-Pooling | $79.67_{(0.23)}$ | $75.08_{(0.29)}$ |

### C.4.3  EFFECT OF DIFFERENT MLP LAYERS

We show performance with different layers of MLP in hypernetwork. Table 10 shows the testing ROC-AUC obtained on SIDER. Here, we constrain that the MLPs in Equation (9)(10)(11) have the same layer number. As shown, using 3 layers reaches the best performance. Please note that although we can set different layer numbers for MLPs used in Equation (9)(10)(11) which further improves performance, setting the same layer number already obtains the state-of-the-art performance. Hence, we set layer number as 3 consistently.

Table 10: Effect of layers of MLP in hypernetwork, testing ROC-AUC obtained on SIDER.

|  | 1 layer | 2 layer | 3 layer | 4 layer |
|---|---|---|---|---|
| 10-shot | $79.98_{(0.35)}$ | $81.85_{(0.33)}$ | $82.40_{(0.26)}$ | $82.43_{(0.28)}$ |
| 1-shot | $75.02_{(0.40)}$ | $76.56_{(0.36)}$ | $77.72_{(0.34)}$ | $77.59_{(0.31)}$ |

### C.5  A CLOSER LOOK AT MOLECULE-LEVEL ADAPTATION

In this section, we pay a closer look at our molecule-level adaptation mechanism, proving evidence of its effectiveness.

### C.5.1  PERFORMANCE UNDER DIFFERENT PROPAGATION DEPTH

Figure 5 compares Pre-PACIA with "w/o M" (introduced in Section 5.3) using different fixed layers of relation graph refinement on Tox21, where the maximum Depth $L = 5$. As can be seen, Pre-PACIA equipped performs much better than "w/o M" which takes the same depth of relation graph refinement as in PAR. This validates the necessity of molecule-level adaptation.

### C.5.2  DISTRIBUTION OF PROPAGATION DEPTH

Figure 6(a) plots the distribution of learned $l'$ for query molecules in meta-testing tasks for 10-shot case of Tox21. The three meta-testing tasks contain different number of query molecules in scale: 6447 in task SR-HSE, 5790 in task SR-MMP, and 6754 in task SR-p53. We can see that Pre-PACIA choose different $tl$ for query molecules in the same task. Besides, the distribution of learned $l'$ varies across different meta-testing tasks: molecules in task SR-MPP mainly choose smaller depth

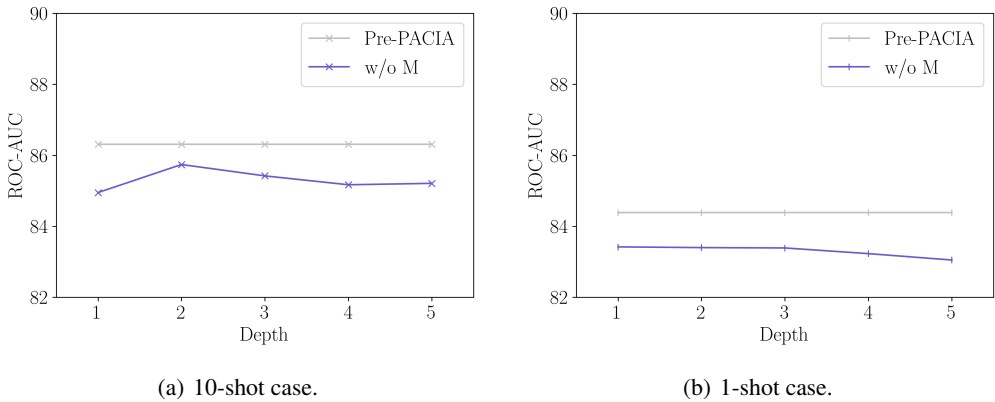

(a) 10-shot case.

(b) 1-shot case.

Figure 5: Comparing Pre-PACIA with "w/o M" using different fixed propagation depth of relation graph on Tox21.

while molecules in the other two tasks tend to choose greater depth. This can be explained as most molecules in task SR-MPP are relatively easy to classify, which is consistent with the fact that Pre-PACIA obtains the highest ROC-AUC on SR-MPP among the three meta-testing tasks (83.75 for SR-HSE, 88.79 for SR-MPP and 86.39 for SR-p53).

Further, we pick out molecules with $l' = 1$ (denote as **Group A**) and $l' = 4$ (denote as **Group B**) as they are more extreme cases. We then apply "w/o M" with different fixed depth for Group A and Group B, and compare them with Pre-PACIA. Figure 6(b) shows the results. Different observations can be made for these two groups. Molecules in Group A have good performance with smaller depth relation graph, they can achieve higher ROC-AUC score than the average of all molecules using Pre-PACIA. These indicate they are easier to classify and it is reasonable that Pre-PACIA choose $l' = 1$ for them. While molecules in Group B are harder to classify and requires $l' = 4$.

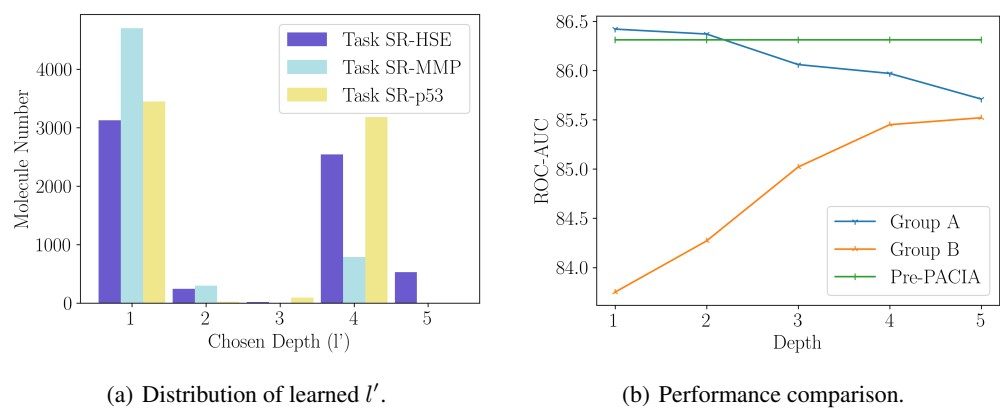

(a) Distribution of learned $l'$.

(b) Performance comparison.

Figure 6: Examine molecule-level adaptation of Pre-PACIA on 10-shot tasks of Tox21.

## C.6 PERFORMANCE GIVEN MORE TRAINING SAMPLES

PACIA can handle general MPP problems. We conduct experiments on SIDER to validate PACIA given increasing labeled samples per task. We compare with GIN (Xu et al., 2019), which is a powerful encoder to handle MPP problems. To make fair comparison, we adapt the commonly used pretraining and fine-tuning strategy. We first pretrain GIN on samples from all meta-training tasks, then use the support set of meta-testing task to fine-tune the classifier. Figure 7 shows the results.

As can be seen, PACIA outperforms GIN for $1, 10, 16, 32, 64$-shot tasks, and is on par with GIN for $128$-shot tasks. The performance gain of PACIA is more significant when fewer labeled samples are provided. Note that all parameters of GNN are fine-tuned, while PACIA only uses a few adaptive parameters to modulate the message passing process of GNN. The empirical evidence shows that PACIA nicely achieves its goal: handling few-shot MPP problem in a parameter-efficient way.

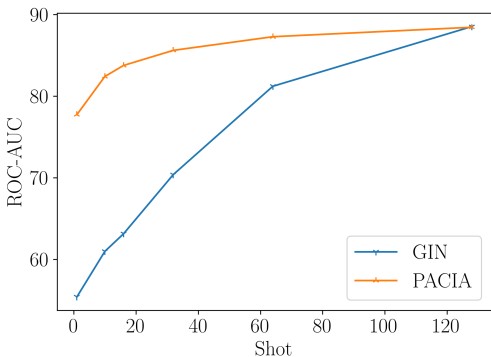

Figure 7: Testing ROC-AUC of PACIA and GIN on SIDER, with different number of labeled samples (shot) per task.

## D ILLUSTRATION FIGURE

Here we provide a more detailed illustration of our proposed PACIA, shown as Figure 8.

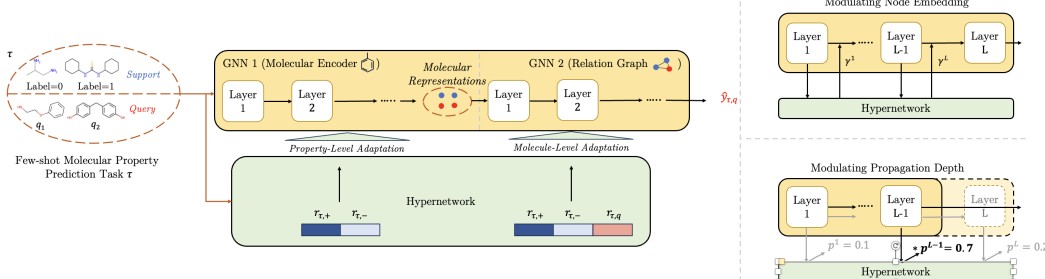

Figure 8: A more detailed illustration of the model of PACIA. The right part shows the proposed unified GNN adapter, which functions by modulating mode embedding and propagation depth of GNN. The left part shows the architecture of PACIA. There are two GNNs in the In the main-network. Hypernetwork perform property-level adaptation on encoder and molecule-level adaptation on predictor.

