# OpenReview forum: "PACIA: Parameter-Efficient Adapter for Few-Shot Molecular Property Prediction"
_ICLR.cc/2024/Conference — Submitted to ICLR 2024_

### Official Review · Reviewer_mPeR · 2023-10-31

**Soundness:** 3 good
**Presentation:** 2 fair
**Contribution:** 2 fair
**Rating:** 3
**Confidence:** 4

**Summary:**

This paper studies the few-shot molecular property prediction problem. Existing gradient-based few-shot methods generally need to update a large number of learnable parameters during the meta-test stage, which is prone to overfitting. To address this problem, this paper proposes the PACIA method the leverages a hypernet to generate adaptive parameters for each task and each molecule in a task.

**Strengths:**

1. This paper studies an important research problem.
2. The proposed method is clear in general and makes sense.

**Weaknesses:**

1. Lack of comparison to SOTA. In Tab. 1 and Tab. 2, the only previous works on few-shot MMP included are ADKF-IFT and PAR. The authors should compare the proposed methods with more existing SOTAs such as [a].
2. Lack of novelty. The core component is HyperNet. Based on HyperNet, the proposed PACIA makes no significant technical contribution.
3. Poor writing. Many sentences are grammatically wrong. Some examples are:
* [In Sec. 1] First is that ... difference.
* [In Sec. 1] The chemical space is enormous that ... range.
* [In Sec. 1] The molecule-level difference ... molecules.
* [In Sec. 1] While others ... accurately.
Note that four of the first 6 sentences of this paragraph are grammatically incorrect.
There are many errors in addition to the above examples. The authors are suggested to carefully proofread the paper to correct the errors.
4. The design in modulating propagation depth seems not fully reasonable. According to Eqn. (8), $[p]_l$ measures the probability of the event that "the $l$-th layer is in the model". However, from Eqn. (7),  the only constraint on $[p]_l $ is $\sum_l [p]_l = 1$. So it is likely that for some $1<i<j<L$, $[p]_i < [p]_j$. That is to say, a hidden layer (i.e., layer $j$) is more likely to be in the model than a layer before it (i.e. layer $i$), which is unreasonable.

**Questions:**

Please see "weakness".

---

> ### Author Response · Authors · 2023-11-18
>
> **Q1. Lack of comparison to SOTA. In Tab. 1 and Tab. 2, the only previous works on few-shot MMP included are ADKF-IFT and PAR. The authors should compare the proposed methods with more existing SOTAs such as [a].**
>
> *Reply*. We compare PACIA with ADKF-IFT and PAR, which are recent baselines obtaining the best performance on MoleculeNet and FS-Mol respectively. We also mention MHNfs (Schimunek et al., 2023), which is not included as it uses additional reference molecules from external datasets.
> For reference [a], could you please provide the paper title?
>
>
> **Q2. Lack of novelty. The core component is HyperNet. Based on HyperNet, the proposed PACIA makes no significant technical contribution.**
>
> *Reply*. Hypernetworks are a general concept involving neural networks that are trained to generate parameters for a main network.
> Designing how a hypernetwork modulates the main network is a problem-specific and non-trivial challenge.
> One cannot directly apply hypernetworks designed for other problems to a new problem.
> Considering the few-shot MPP problem, existing works can be summarized within an encoder-predictor framework, where GNNs perform effectively, acting as both encoders and predictors.
> Correspondingly, we have designed a unified GNN adapter that generates a few adaptive parameters. These parameters modulate the message passing process of the GNN in two key aspects: node embedding and propagation depth.
> As a result, the encoder adapts at the property level, while the predictor adapts at the molecule level.
> Our proposed PACIA effectively addresses the few-shot MPP problem, achieving state-of-the-art performance on benchmark datasets.
> In conclusion, PACIA contributes significantly to both the technological and empirical advancements in the field of few-shot MPP.
>
> **Q3. Poor writing. Many sentences are grammatically wrong.**
>
> *Reply*. We sincerely apologize for the grammatical issues. We have now carefully proofread the manuscript and marked the grammar corrections in the revised version.
>
> **Q4. The design in modulating propagation depth seems not fully reasonable. The design in modulating propagation depth seems not fully reasonable. According to Eqn. (8), $[p]_l$ measures the probability of the event that "the $l$-th layer is in the model". However, from Eqn. (7), the only constraint on $[p]_l$ is $\sum_l [p]_l=1$. So it is likely that for some $1<i<j<L$, $[p]_i<[p]_j$. That is to say, a hidden layer (i.e., layer $j$) is more likely to be in the model than a layer before it (i.e. layer $i$), which is unreasonable.**
>
> *Reply*. This is a misunderstanding: the relative magnitude between $[p]_i$ and $[p]_j$ is independent of the order between i and j.
>
> For any input sample $\mathcal{X}_{\tau,q}$, it will always go through a complete message passing procedure in a $L$-layer GNN. Afterwards, for each layer $l\in1,2\cdots,L$, hypernetwork takes the node embeddings after $l$-th layer (i.e., {$\textbf{h}^l$}) as input, and generates a scalar $[p]_l$ representing how likely it can get good performance by using {$\textbf{h}^l$} as the final node embeddings.
> During meta-training, $\{[p]_l\}$ act as differentiable combination weights in Equation (8). In this way, the hypernetwork is trained to learn the optimal mapping from {$\textbf{h}^l$} to $[p]_l$, through gradient flow among Equation (12)-(3)-(5)-(8)-(10) or Equation (12)-(3)-(5)-(8)-(11).
> During meta-testing, we directly select one optimal layer $l'$ by Equation (13). Then, only {$\textbf{h}^{l'}$} of $l'$th layer are fed to next module.

---

### Official Review · Reviewer_smzw · 2023-11-01

**Soundness:** 3 good
**Presentation:** 3 good
**Contribution:** 3 good
**Rating:** 6
**Confidence:** 3

**Summary:**

PACIA is a novel approach aimed at addressing the challenges in Molecular Property Prediction (MPP) when labeled data is scarce. The authors identify that existing methods, which typically rely on a gradient-based strategy for property-level adaptation, are prone to overfitting due to the large number of adaptive parameters required. To overcome this, they introduce PACIA, a parameter-efficient Graph Neural Network (GNN) adapter specifically designed for few-shot MPP scenarios.

**Strengths:**

Innovative Solution: PACIA introduces a novel parameter-efficient adapter for Few-Shot Molecular Property Prediction (MPP), addressing the challenge of overfitting in scenarios with scarce labeled data. This approach stands out due to its unique application of a hierarchical adaptation mechanism, modulating both the encoder and predictor in a GNN framework.

Well-Defined Problem and Solution: The paper clearly defines the problem of few-shot MPP and presents PACIA as a well-justified solution. The hierarchical adaptation mechanism is meticulously designed, reflecting the high quality of the work.

Advancement in Methodology: The introduction of a parameter-efficient adapter and the application of hierarchical adaptation in GNNs for MPP represent a notable advancement in methodology, setting a precedent for future work in the domain.

**Weaknesses:**

Lack of Ablation Studies on Hypernetworks:
While the paper introduces the innovative use of hypernetworks for generating adaptive parameters, it lacks ablation studies or a deeper analysis of how different configurations of hypernetworks affect the performance of PACIA. Incorporating ablation studies or a detailed analysis focused on the hypernetworks component would provide valuable insights into its role and optimization, potentially leading to further improvements in PACIA’s performance.

Need for Broader Applicability and Generalization:
The paper validates PACIA’s performance in few-shot MPP problems, but it could strengthen its case by demonstrating the model’s applicability and generalization across a wider range of molecular property prediction tasks. Conducting experiments or providing examples of PACIA’s performance in diverse MPP tasks would showcase its versatility and generalization capabilities, further solidifying its contributions to the field.

**Questions:**

Comprehensive Comparison with Baselines: Could the authors provide additional comparisons with a broader range of existing methods, especially those that have shown promising results in related domains, to strengthen the validation of PACIA’s performance?

Analysis of Hypernetworks: How do different configurations or architectures of hypernetworks affect the performance of PACIA? Are there specific settings that are more optimal for this application?

---

> ### Author Response · Authors · 2023-11-18
>
> **Q1. Comprehensive Comparison with Baselines: Could the authors provide additional comparisons with a broader range of existing methods, especially those that have shown promising results in related domains, to strengthen the validation of PACIA’s performance?**
>
> *Reply*. Please check Q2 in global response.
>
>
> **Q2. Analysis of Hypernetworks: How do different configurations or architectures of hypernetworks affect the performance of PACIA? Are there specific settings that are more optimal for this application?**
>
> *Reply*.
> For specific settings, please refer to Table 4 and Table 5 for hyperparameters chosen on MoleculeNet and FS-Mol, where the configuration of hypernetwork is included.
>
> For ablation studies, we now add results concerning three aspects in hypernetwork in Appendix C.4, which are reproduced below.
>
> 1.Effect of concatenating label $\textbf{y}_{{\tau},s}$ in Equation (9). Table below shows the testing ROC-AUC obtained on SIDER. As shown, "w/ Label" helps keep the label information in support set, which improves the performance.
>
> |     | 10-shot  | 1-shot    |
> | -------- | -------- | -------- |
> | w/ Label| $82.40_{(0.26)}$ | $77.72_{(0.34)}$ |
> | w/o Label | $76.91_{(0.17)}$ | $74.10_{(0.41)}$|
>
>
> 2.Different ways of combining active prototype $\textbf{r}^l_{\tau,+}$ and inactive prototype $\textbf{r}_{{\tau},-}^l$ in Equation (10) and Equation (11).
> As "Concatenating" active and inactive prototypes allows MLP to capture more complex patterns, it obtains better performance on SIDER as shown in the table below.
>
> |    | 10-shot  | 1-shot    |
> | -------- | -------- | -------- |
> | Concatenating| $82.40_{(0.26)}$ | $77.72_{(0.34)}$ |
> | Mean-Pooling | $79.67_{(0.23)}$ | $75.08_{(0.29)}$ |
>
> 3.Effect of layers of MLP in hypernetwork.
> Table below shows the testing ROC-AUC obtained on SIDER.
> Here, we constrain that the MLPs in Equation (9-11) have the same layer number.
> As shown, using 3 layers reaches the best performance.
> Please note that although we can set different layer numbers for MLPs used in Equation (9-11) which further improves performance, setting the same layer number already obtains the state-of-the-art performance. Hence, we set layer number as 3 consistently.
>
> |     |  1 layer | 2 layer  | 3 layer  | 4 layer  |
> | -------- | -------- | -------- | -------- | -------- |
> | 10-shot| $79.98_{(0.35)}$ | $81.85_{(0.33)}$ |$82.40_{(0.26)}$ |$82.43_{(0.28)}$ |
> | 1-shot   | $75.02_{(0.40)}$ | $76.56_{(0.36)}$ |$77.72_{(0.34)}$ |$77.59_{(0.31)}$ |

---

### Official Review · Reviewer_WZaL · 2023-11-04

**Soundness:** 3 good
**Presentation:** 3 good
**Contribution:** 2 fair
**Rating:** 6
**Confidence:** 2

**Summary:**

The paper proposes a parameter-efficient approach for few-shot molecular property prediction (MPP) tasks by involving hypernetwork to modulate the GNN parameters. The proposed PACIA is built on top of a main encoder-decoder MPP network, and by learning from the training, a GNN adapter is trained to modulate the node embedding and GNN depth. Extensive experiments are conducted in two settings to evaluate the performance of PACIA. Several in-depth analysis is provided to further discuss the superiority of PACIA, such as running time.

**Strengths:**

+ The paper is interesting as it presents another direction for few-shot MPP. Unlike general gradient-based approaches, PACIA tends to learn certain key generalized parameters to minimize the training costs.
+ The paper is well-written and easy to follow.
+ The authors have conducted several in-depth analysis to comprehensively evaluate the performance of the proposed method.

**Weaknesses:**

- The approaches of modulating node embedding and GNN depth do not have sufficient theoretical support. It is more like experimental attempts. Can the authors provide more details about why the implementation is designed as such? How does such implementation ensure the adaptor learns sufficient information?
- The main framework of PACIA is based on PAR, making the technical novelty incremental.
- The figure font is small and hard to recognize. Fig.1 (b) is too abstract. The authors may consider plotting a more detailed overall framework to help understand their method.
- In Table 2, why the baseline methods are different? PAR is the most similar baseline model, and should be compared.

**Questions:**

See Weaknesses.

The proposed method is interesting and has certain merit. I am also curious that will it works on general MPP problems? Did the authors try to see the performance not under the few-shot setting?

---

> ### Author Response · Authors · 2023-11-18
>
> **Q1. The approaches of modulating node embedding and GNN depth do not have sufficient theoretical support. It is more like experimental attempts. Can the authors provide more details about why the implementation is designed as such? How does such implementation ensure the adaptor learns sufficient information?**
>
> *Reply*. Our PACIA is designed upon hypernetworks. It is proved that hypernetworks can be more expressive and can reach the same error with lower complexity (number of trainable parameters) than embedding-based methods in general [a1]. A line of amortization-based approaches (Requeima et al., 2019; Lin et al., 2021; Przewiezlikowski et al., 2022) have shown the effectiveness of designing hypernetworks to handle few-shot image classification and cold-start recommendation. We share the same spirit. While theoretically analyzing a particular hypernetwork is still an open question, we empirically analyze the contribution of each component in PACIA in Section 5.3, and provide evidence of property-level adaptation and molecule-level adaptation in Section 5.4.
>
> [a1] Galanti, T. and Wolf, L. (2020). On the modularity of hypernetworks. Advances in Neural Information Processing Systems, 33, 10409-10419.
>
> **Q2. The main framework of PACIA is based on PAR, making the technical novelty incremental.**
>
> *Reply*. Please note that we propose PACIA to enhance the encoder-predictor framework, which is adopted by existing works, with a parameter-efficient GNN adapter, rather than merely aiming to improve PAR.
> As discussed in Section 3.2,
> existing works take GNN as molecular encoder consistently, and predict by pair-wise similarity (Altae-Tran et al., 2017), multi-layer perceptron (Guo et al., 2021), Mahalanobis distance (Stanley et al., 2021), and GNN operates on relation graph (Wang et al., 2021).
> In particular, PACIA learns to generate a few adaptive parameters to modulate the message passing process of GNN.
> If it is only applied on GNN encoder, i.e., w/o M variant in Section 5.3, it obtains 83.62 and 82.01 for 10-shot and 1-shot tasks of Tox21. These results already outperform the best results obtained by existing methods, i.e.,  82.43 obtained by ADKF-IFT and 80.02 obtained by PAR for 10-shot and 1-shot tasks respectively.
> Further, as the recent PAR obtains the state-of-the-art performance by learning a GNN on relation graphs, we can apply our adapter to modulate this GNN predictor on a molecular level. The benefit is that molecular-level difference can be captured. Empirical results in Figure 2(a) and case study in Section 5.4 validates that applying molecule-level adaptation is useful.
>
>
> **Q3. The figure font is small and hard to recognize. Fig.1 (b) is too abstract. The authors may consider plotting a more detailed overall framework to help understand their method.**
>
> *Reply*. We are sorry for the inconvenience. We have revised Fig.1 (b) and added a more detailed one in Appendix D.
>
> **Q4. In Table 2, why the baseline methods are different? PAR is the most similar baseline model, and should be compared.**
>
> *Reply*. Please check Q1 in global response.
>
> **Q5. I am also curious that will it works on general MPP problems? Did the authors try to see the performance not under the few-shot setting?**
>
> *Reply*. Please check Q2 in global response.

---

> > ### Comment · Reviewer_WZaL · 2023-11-22
> >
> > Thank the authors for the rebuttal. I will consider updating my score after discussing with other reviewers.

---

### Official Review · Reviewer_tnxf · 2023-11-06

**Soundness:** 3 good
**Presentation:** 2 fair
**Contribution:** 2 fair
**Rating:** 5
**Confidence:** 4

**Summary:**

This paper delves into the challenges of few-shot molecular property prediction, highlighting two major limitations in current approaches: the neglect of molecule-level differences and a predisposition to overfitting. In response, the authors introduce a parameter-efficient adapter complemented by a molecule-adaptive predictor. The experimental results on various benchmark datasets have demonstrated the effectiveness of the proposed method.

**Strengths:**

1.	The paper focuses on an intriguing and pivotal issue. The scarcity of labeled datasets is a prevalent challenge in the realm of chemistry.
2.	The proposed method is well-motivated. The introduction lucidly underscores the drawbacks of the existing works, and each module designed in this study directly addresses these shortcomings.
3.	The experimental results shown in Table 1 and Table 2 clearly demonstrate the effectiveness of the proposed method compared with the various baselines. Additionally, the authors have undertaken an exhaustive ablation study that accentuates the significance of each individual component.

**Weaknesses:**

1.	The exposition on the methodology appears somewhat nebulous, which hampers a clear comprehension of the distinct contributions of each module. Specifically, the average representation at the l-th GNN layer, as depicted in Equation (9), seems disconnected from subsequent steps. The property adaptation and molecule adaptation are not clear in the algorithm.
2.	There is a noticeable discrepancy in the baselines used for comparison in Tables 1 and 2. The rationale behind this difference remains unexplained. For instance, while PAR[1] is conspicuously absent from Table 2, it seems like a plausible candidate for few-shot molecular property prediction in FS-Mol.
3.	The presentation of results in Tables 1 and 2 would benefit from a consistent format, ensuring ease of interpretation for readers.


[1] Property-aware relation networks for few-shot molecular property prediction. NeurIPS 2021.

**Questions:**

Please refer to the weaknesses

---

> ### Author Response · Authors · 2023-11-18
>
> **Q1. The exposition on the methodology appears somewhat nebulous, which hampers a clear comprehension of the distinct contributions of each module. Specifically, the average representation at the l-th GNN layer, as depicted in Equation (9), seems disconnected from subsequent steps. The property adaptation and molecule adaptation are not clear in the algorithm.**
>
> *Reply*. We apologize for the confusion.
> We have revised Generating Adaptive Parameters by Hypernetwork paragraph in Section 4.1 to make it clearer.
> Property-level adaptation is generated by Equation (9) and Equation (10), and molecule-level adaptation is generated by Equation (9) and Equation (11).
> In Equation (9), we obtain class prototypes $\textbf{r}\_{{\tau},+}^l$ and $\textbf{r}\_{{\tau},-}^l$ by mean-pooling over the embeddings of samples in active class (+) and inactive class (-) from $\mathcal{S}\_{\tau}$. Consequently, the generated adaptive parameter are permutation-invariant to the order of input samples in $\mathcal{S}_{\tau}$.
>
> **Q2. There is a noticeable discrepancy in the baselines used for comparison in Tables 1 and 2. The rationale behind this difference remains unexplained. For instance, while PAR[1] is conspicuously absent from Table 2, it seems like a plausible candidate for few-shot molecular property prediction in FS-Mol.**
>
> *Reply*. Please check Q1 in global response.
>
> **Q3. The presentation of results in Tables 1 and 2 would benefit from a consistent format, ensuring ease of interpretation for readers.**
>
> *Reply*. Thanks for the advice. We now use a consistent format for Table 1 and Table 2 in the revised manuscript.

---

### Author Response · Authors · 2023-11-18
**Global Response Q2**

**Q2. Will it work on general MPP problems? Did the authors try to see the performance not under the few-shot setting?**

*Reply*.
PACIA can handle general MPP problems. We conduct experiments on SIDER to validate PACIA given increasing labeled samples per task. We compare PACIA with GIN (Xu et al., 2019), which is a powerful encoder to handle MPP problems. To make a fair comparison, we adapt the commonly used pretraining and fine-tuning strategy. We first pretrain GIN on samples from all meta-training tasks, then use the support set of meta-testing task to fine-tune the classifier.
The results are reported in Figure 7 in the Appendix C.6 and reproduced as Table below.
As can be seen, PACIA outperforms GIN for $1, 10, 16, 32, 64$-shot tasks, and is on par with GIN for $128$-shot tasks.
The performance gain of PACIA is more significant when fewer labeled samples are provided.
Note that all parameters of GNN are fine-tuned, while PACIA only uses a few adaptive parameters to modulate the message passing process of GNN.
The empirical evidence shows that PACIA nicely achieves its goal: handling few-shot MPP problem in a parameter-efficient way.

|     | 1-shot  | 10-shot    | 16-shot    | 32-shot    | 64-shot   | 128-shot   |
| -------- | -------- | -------- | -------- | -------- | -------- | -------- |
| GIN |$55.39_{(1.83)}$ | $60.97_{(1.02)}$ |$63.12_{(1.09)}$ |$70.38_{(0.94)}$ |$81.20_{(0.88)}$ |$88.52_{(0.63)}$
| PACIA | ${77.72_{(0.34)}}$| ${82.40_{(0.26)}}$ |$83.75_{(0.46)}$ |$85.60_{(0.33)}$ |$87.27_{(0.29)}$ |$88.53_{(0.30)}$

---

### Author Response · Authors · 2023-11-18
**Global Response Q1**

**Q1. Why use different baselines for comparison in Tables 1 and 2?**

*Reply*.
In the submitted manuscript,
we adopt different baseline sets for MoleculeNet and FS-Mol as these two datasets have large discrepancy.
As discussed in Setting paragraphs of Section 5.1 and Section 5.2, MoleculeNet is commonly used to evaluate the performance of $2$-way $K$-shot few-shot MPP problem where both inactive and active classes are provided with $K$ labeled samples. While FS-Mol contains more diverse tasks, and its classes are imbalanced.
Therefore, different baselines are proposed to handle these two datasets, i.g. (Altae-Tran et al., 2017; Guo et al., 2021; Wang et al., 2021) for MoleculeNet and (Stanleyetal.,2021; Schimunek et al., 2023) for FS-Mol.
While the recent proposed (Chen et al., 2022) and our PACIA can handle both settings, PACIA obtains the best performance.

As suggested by the reviewers, we manage to align these baselines and compare PACIA with the same set of baselines on both datasets.
Please see the updated Table 1 and 2 (reproduced below) in the revised manuscript for results. As can be seen, PACIA still performs the best.

Table 1: Test ROC-AUC obtained on MoleculeNet. The best results are bolded, second-best results are underlined. Random Forest is not applicable as it requires fingerprints which are not available in MoleculeNet.

|     | Tox21 (10-shot)    | Tox21 (1-shot)    | SIDER (10-shot)    | SIDER(1-shot)    | MUV (10-shot)    | MUV (1-shot)    | ToxCast (10-shot)    | ToxCast (1-shot)    |
| -------- | -------- | -------- | -------- | -------- | -------- | -------- | -------- | -------- |
| Random Forest| - | - | - | - | - | - | - | - |
| GNN-ST | $61.23_{(0.89)}$|$55.49_{(2.31)}$|$56.25_{(1.50)}$|$52.98_{(2.12)}$|$54.26_{(3.61)}$|$51.42_{(5.11)}$|$55.66_{(1.47)}$|$51.80_{(1.99)}$|
| MAT | $64.84_{(0.93)}$|$54.90_{(1.89)}$|$57.45_{(1.26)}$|$52.97_{(3.00)}$|$56.19_{(2.88)}$|$52.01_{(4.05)}$|$58.50_{(1.62)}$|$52.41_{(2.34)}$|
| GNN-MT | $69.56_{(1.10)}$|$62.08_{(1.25)}$|$60.97_{(1.02)}$|$55.39_{(1.83)}$|$66.24_{(2.40)}$|$60.78_{(2.91)}$|$65.72_{(1.19)}$|$62.38_{(1.67)}$|
|ProtoNet|$72.99_{(0.56)}$|$68.22_{(0.46)}$|$61.34_{(1.08)}$|$57.41_{(0.76)}$|$68.92_{(1.64)}$|$64.81_{(1.95)}$|$65.29_{(0.82)}$|$63.73_{(1.18)}$
|MAML|$79.59_{(0.33)}$|$75.63_{(0.18)}$|$70.49_{(0.54)}$|$68.63_{(1.51)}$|$68.38_{(1.27)}$|$65.82_{(2.49)}$|$68.43_{(1.85)}$|$66.75_{(1.62)}$
|Siamese|$80.40_{(0.29)}$|$65.00_{(11.69)}$|$71.10_{(1.68)}$|$51.43_{(2.83)}$|$59.96_{(3.56)}$|$50.00_{(0.19)}$|-|-
|EGNN|$80.11_{(0.31)}$|$75.71_{(0.21)}$|$71.24_{(0.37)}$|$66.36_{(0.29)}$|$68.84_{(1.35)}$|$62.72_{(1.97)}$|$66.42_{(0.77)}$|$63.98_{(1.20)}$
|IterRefLSTM|$81.10_{(0.10)}$|$80.97_{(0.06)}$|$69.63_{(0.16)}$|$71.73_{(0.06)}$|$49.56_{(2.32)}$|$48.54_{(1.48)}$|-|-
|PAR|$82.13_{(0.26)}$|$\underline{80.02}_{(0.30)}$|$\underline{75.15}_{(0.35)}$|$\underline{72.33}_{(0.47)}$|$68.08_{(2.23)}$|$65.62_{(3.49)}$|$70.01_{(0.85)}$|$\underline{68.22}_{(1.34)}$
|ADKF-IFT|$\underline{82.43}_{(0.60)}$|$77.94_{(0.91)}$|$67.72_{(1.21)}$|$58.69_{(1.44)}$|$\textbf{98.18}_{(3.05)}$|$\underline{67.04}_{(4.86)}$|$\underline{72.07}_{(0.81)}$|$67.50_{(1.23)}$
|PACIA (ours)|$\textbf{84.25}_{(0.31)}$|$\textbf{82.77}_{(0.15)}$|$\textbf{82.40}_{(0.26)}$|$\textbf{77.72}_{(0.34)}$|$\underline{72.58}_{(2.23)}$|$\textbf{68.80}_{(4.01)}$|$\textbf{72.38}_{(0.96)}$|$\textbf{69.89}_{(1.17)}$

Table 2: Test $\Delta$ AUPRC obtained on FS-Mol. Tasks are categorized by target protein type and the number of tasks per category is reported in brackets. The best results are bolded, second-best results are underlined. The results of Random Forest, Single-Task GNN, MAT, Multi-Task GNN, MAML, PAR, ProtoNet, Siamese, IterRefLSTM, ADKF-IFT are copied from (Schimunek et al., 2023).

|     | All[157]  | Kinases[125] | Hydrolases[20]  | Oxidoreductases[7] |
| -------- | -------- | -------- | -------- | -------- |
|Random Forest|$.092 _{(.007)}$|$.081 _{( .009)}$|$.158 _{( .028)}$|$.080 _{( .029)}$
|GNN-ST|$.029 _{( .004)}$|$.027 _{( .004)}$|$.040 _{( .018)}$|$.020 _{( .016)}$
|MAT|$.052 _{( .005)}$|$.043 _{( .005)}$|$.095 _{( .019)}$|$.062 _{( .024)}$
|GNN-MT|$.093 _{( .006)}$|$.093 _{( .006)}$|$.108 _{( .025)}$|$.053 _{( .018)}$
|MAML|$.159 _{( .009)}$|$.177 _{( .009)}$|$.105 _{( .024)}$|$.054 _{( .028)}$
|PAR| $.164_{(.008)}$ | $.182_{(.009)}$ | $.109_{(.020)}$ | $.039_{(.008)}$ |
|ProtoNet|$.207 _{( .008)}$|$.215 _{( .009)}$|$.209 _{( .030)}$|$.095 _{( .029)}$
|EGNN |$.212_{(.011)}$|$.224_{(.010)}$ |$.205_{(.024)}$|$.097_{(.022)}$
|Siamese |$.223_{(.010)}$ | $.241_{(.010)}$ | $.178_{(.026)}$ | $.082_{(.025)}$ |
|IterRefLSTM |$\underline{.234}_{(.010)}$ | $\textbf{.251}_{(.010)}$ | $.199_{(.026)}$ | $.098_{(.027)}$ |
|ADKF-IFT|$\underline{.234} _{( .009)}$|${.248} _{( .020)}$|${\underline{.217 }}_{( .017)}$|$\textbf{.106} _{( .008)}$
|PACIA (ours)|$\textbf{.236}_{( .008)}$|$\textbf{.251} _{( .016)}$|$\textbf{.219}_{( .029)}$|$\textbf{.106} _{( .010)}$

---

### Author Response · Authors · 2023-11-18
**Global Response**

We sincerely thank all reviewers for their valuable comments. We have revised the uploaded PDF, where any contents that differ from the first submission version are highlighted in blue.

---

### Meta-Review · Area_Chair_ppR2 · 2023-12-06

**Metareview:**

The paper introduces a novel method for few shot learning in molecule property prediction. It is an important area that requires further investigation. The labels are often scarce in chemical tasks which necessitates developing methods that can work with very few annotations.

Reviewers have not reached an agreement.  I would like also to add that I have downweighted the review by mPeR in the final decision, as the Reviewer has not engaged in the rebuttal process, while it was necessary due to the breviety of the comments.

Among the significant concerns raised by reviewers were: (a) a lack of clarity on the exposition and motivation of the method, (b) a lack of technical novelty. There was also a concern with not including certain models, which was addressed in the discussion phase.

I agree with the first concern, but I also think that this aspect has been improved during the discussion phase. Even so, the paper is still unclear at times, e.g. when the Authors claim that “molecule-level differences” are not addressed by other few shot learning methods.

The second concern was the most important in the decision. Indeed the work is a relatively straightforward adaptation of adapters/hypernetworks to the task. The performance is somewhat comparable to the next best-performing method (ADKF-IFT). From this perspective, this work might be of limited interest to the ICLR community, while might be a very valuable addition to journals less focused on novelty or topical journals in chemistry.

All in all, I am unfortunately recommending at this stage rejection. I would like to thank the Authors for the submission and I hope comments will be helpful in improving the work.

**Justification For Why Not Higher Score:**

The main concern is technological novelty.

**Justification For Why Not Lower Score:**

N/A

---

### Decision · Program_Chairs · 2024-01-16

Reject